# Identification of prognostic genes associated with tolerogenic dendritic cells in gastric cancer based on transcriptomic data

**Shenyu Luo, Guozhi Yang, Yuhua Yuan, Yong Zhang, Yihua Kang, Zhengrong Wen, Wei Liu, Ying Liu** [ID]*

The People's Hospital of Chuxiong Yi Autonomous Prefecture, Chuxiong, Yunnan, China

* yaningliu27@qq.com

## Abstract

### Background

Tolerogenic dendritic cells have a pivotal function in treating autoimmune illnesses, atopic diseases, and neoplasms. The precise mechanism by which Tolerogenic dendritic cells function in gastric cancer remains incompletely understood. Therefore, this research explored potential genes with prognostic value related to Tolerogenic dendritic cells in gastric cancer, to identify novel therapeutic targets that could provide valuable insights for the clinical treatment of gastric cancer.

### Results

Five prognostic genes (CXCL1, INHBA, ASCL2, RNASE1, and GPX3) were finally obtained to construct the risk model. Immune infiltration analysis revealed that GPX3 exhibited significant positive associations with various immune cell populations, particularly regulatory T cells. While ASCL2 was weakly associated with almost all immune cells. These results suggested that there was a complex correlation between prognostic genes and immune cells. The analysis of drug sensitivity demonstrated higher IC50 values for compounds such as BIBW2992 in high-risk group relative to low-risk group. A reverse pattern was observed for GSK269962A and similar drugs, which showed significantly higher IC50 values in low-risk group than high-risk group.

### Conclusion

The present study revealed five prognostic genes and constructed a predictive model, which provided a theoretical basis for the correlation linking Tolerogenic dendritic cells to gastric cancer, and established potential therapeutic strategies in managing gastric cancer. Single-cell analysis revealed that INHBA, ASCL2, and CD36 exhibited marked differential expression in dendritic cells.

**Data availability statement:** The datasets analysed in this study are available in the Gene Expression Omnibus (GEO) database (http://www.ncbi.nlm.nih.gov/geo/), including GSE66229 dataset, GSE23371 datase, GSE52894 dataset, GSE56017 dataset, and GSE182528 dataset, and he Cancer Genome Atlas database (https://portal.gdc.cancer.gov/), including TCGA-STAD dataset. All relevant data are within the manuscript and its Supporting information files.

**Funding:** The author(s) received no specific funding for this work.

**Competing interests:** The authors have declared that no competing interests exist.

**Abbreviations:** tolDCs, Tolerogenic Dendritic Cells; GC, Gastric Cancer; HRG, High-Risk Group; LRG, Low-Risk Group; m6A, N6-methyladenosine; ZFPs, Zinc Finger Proteins; DCs, Dendritic Cells; KEGG, Kyoto Encyclopedia of Genes and Genomes; Tregs, T Cells; MBC, Metastatic Breast Cancer; TME, Tumor Microenvironment; ICB, Immune Checkpoint Inhibitors; BC, Breast Cancer; LPS, Lipopolysaccharide; DEGs, Differentially Expressed Genes; GSEA, Gene Set Enrichment Analysis; FC, Fold Change; KEGG, Kyoto Encyclopedia of Genes and Genomes; GO, Gene Ontology; BP, Biological Process; CC, Cellular Component; MF, Molecular Function; STRING, Search Tool for the Retrieval of Interacting Genes; PPI, Protein-Protein Interaction; HR, Hazard Ratio; PH, Proportional Hazards; LASSO, Least Absolute Shrinkage and Selection Operator; ROC, Receiver Operating Characteristic; AUC, Area Under the Curve; miRNA, MicroRNA; MSigDB, Molecular Signatures Database; NES, Normalized Enrichment Score; GDSC, Genomics of Drug Sensitivity in Cancer; CXCL1, C-X-C motif Ligand 1; GRO-α, Growth-Regulated Gene Alpha; CXCR2, C-X-C Motif Receptor 2; ICIs, Immune Checkpoint Inhibitors; hGC-MSCs, Human GC-Originated Mesenchymal Stem Cells; INHBA, Inhibin Subunit Beta A; TGF-β, Transforming Growth Factor-Beta; AEG, Adenocarcinomas of the Esophagus; AS, Stomach; ASCL2, Achaete-Scute Family bHLH Transcription Factor 2; bHLH, Basic Helix-Loop-Helix; CSCs, Cancer Stem Cells; GPX3, Glutathione Peroxidase 3; RNASE2, Ribonuclease A Family Member; EDN, Eosinophil-Derived Neurotoxin; SLE, Systemic Lupus Erythematosus; ABCs, Age-Associated B Cells; PKP2, Plakophilin-2.

## 1. Introduction

According to the GLOBOCAN cancer statistics, gastric cancer (GC) ranked fifth in both incidence and mortality across cancers. there will be more than 1.1 million new cases of GC and 770,000 deaths in 2022, accounting for 43% of new cases from China [1]. The risk factors for GC encompass age, sex, obesity and metabolic disorders, alcohol consumption, medications, genetic factors, and Helicobacter pylori or other microbial factors [2]. Furthermore, recent studies have revealed more molecular mechanisms of GC, such as gastric microecological dysbiosis is closely associated with the onset and progression of GC [3], N6-methyladenosine (m6A) methylation can affect the growth and progression of GC through immunomodulation [4], zinc finger proteins (ZFPs) can affect the cell proliferation, invasion and metastasis, immune invasion, and drug resistance of gastric cancer [5]. Despite the significant advances that have been made in the study of molecular mechanisms of GC, many challenges, including that the molecular mechanism of gastric cancer has not been systematically elucidated and the biomarkers with definitive cut-off values are still unknown, remain and hinder the early diagnosis and treatment of GC. Therefore, the identification of reliable diagnostic markers and predictive biomarkers plays a crucial role in the clinical management of gastric cancer. Dendritic cells (DCs) play a key role in modulating both innate and adaptive immunity. Tolerogenic dendritic cells (tolDCs), a specialized subset of DCs resistant to full maturation, are capable of inducing regulatory immune responses, can induce immune tolerance. The tolDCs promote central and peripheral tolerance via regulating the expression of immunomodulatory molecules, mediating the inactivation and apoptosis of T cells, also promoting the generation of regulatory T cells (Tregs). The tolDCs can promote the progression of multiple tumors and are linked to poor prognosis of patients [6]. Studies indicate that TolDCs contribute to the aggressive progression of breast cancer (BC) by supporting tumor cell proliferation, invasion, migration, neovascularization, and metastasis, thereby increasing the likelihood of poor outcomes and higher death rates in metastatic breast cancer (MBC) patients [7]. In colorectal cancer, the presence of an immunosuppressive tumor microenvironment (TME) induces tolDCs from immunostimulatory into tolerogenic and immunosuppressive phenotypes and functions, affecting T cells' survival and proliferation, contributing to immune escape and dampening of anti-tumor T cells responses [8]. The tolDCs can affect the occurrence and development of lung cancer by inhibiting the role of immune checkpoint inhibitors (ICB) [9]. Otherwise, the tolDCs play a critical role in many immune-related diseases via promoting immunological tolerance and immunosuppression through various mechanisms, which include regulating T-cell differentiation [10], depletion of autoreactive T cells, expression of inhibitory molecules, and peripheral induction of Tregs [11,12]. Immune evasion in gastric cancer is intricately linked to the characteristics and cellular interactions within the tumor microenvironment; the exhaustion and functional inhibition of the immune cells that possess tumor-suppressive capabilities play a central role in the development of tumor-associated immune tolerance and immune escape [13]. The role of immune tolerance and immunosuppression of tolDCs in GC remains unclear and needs further study.

This study explored the potential prognostic genes related to GC and tolDC based on TCGA and GEO public databases through bioinformatics methods. A risk prognostic model is constructed based on GC-tolDC-related prognostic genes, and the molecular regulatory mechanisms involved in prognostic genes are explored in depth, which is helpful to further reveal the relationship between GC and tolDCs. Our study is expected to provide a new reference for targeted therapy and clinical diagnosis of GC patients, and provide a new direction for immunotherapy research of GC by integrating the genes related to tolDCs in the differentially expressed genes in GC and associated with the prognosis of GC patients.

## 2. Materials and methods

### 2.1. Data collection

The TCGA-STAD dataset functioned as the training cohort, providing transcriptomic, prognostic, and clinical data for gastric cancer (GC) from The Cancer Genome Atlas database (https://portal.gdc.cancer.gov/) (accessed March 20th, 2025). From TCGA-STAD, 448 samples were collected, including 420 tumor tissue samples from GC (361 samples with survival information) and 35 normal tissue samples [14]. In addition, the GSE66229 dataset (GPL570 platform) was downloaded from the Gene Expression Omnibus (GEO, https://www.ncbi.nlm.nih.gov/geo/) to serve as the validation set. This dataset comprised 299 gastric cancer tissue samples with corresponding survival information [14].

To systematically identify tolerogenic dendritic cell-related genes, we further collected four independent tolDCs datasets from the GEO database:

tolDCs dataset 1 (GSE23371, GPL570 platform) comprised 3 samples of activated tolerogenic monocyte-derived dendritic cells (moDCs) (experimental group) and 3 control tolerogenic moDCs samples.

tolDCs dataset 2 (GSE52894, GPL10558 platform) comprised 5 lipopolysaccharide-tolerized moDCs samples (experimental group) and 4 control tolerogenic moDCs samples.

tolDCs dataset 3 (GSE56017, GPL570 platform) comprised 8 tolerogenic moDCs samples (experimental group) and 7 mature moDCs samples (control group).

tolDCs dataset 4 (GSE182528, GPL570 platform) comprised 5 samples of in vitro-induced dexamethasone-tolerized dendritic cells (experimental group) and 5 samples of in vitro-induced moDCs (control group).

Finally, the single-cell RNA sequencing dataset GSE183904 (Platform: GPL24676) was obtained from the GEO database. This dataset comprised 36 samples, including 10 normal gastric tissues and 26 gastric cancer tissues. All samples were included in the single-cell analysis for this study. The data access date was October 14, 2025.

### 2.2. Differential expression analysis in the tolDCs-related genes (tolDCs-RGs)

To systematically identify genes significantly associated with tolerogenic dendritic cells (tolDCs), differential expression analysis was performed across the four independent tolDCs datasets. The R package "limma" [15] was utilized to identify differentially expressed genes (DEGs) between the experimental and control groups in each dataset. The filtering criteria were set at an absolute log2 fold change (FC) > 2 and an adjusted P-value < 0.01. This analysis sequentially identified four distinct gene sets: tolDCs-DEGs1, tolDCs-DEGs2, tolDCs-DEGs3, and tolDCs-DEGs4. Subsequently, these four sets of DEGs were merged, and duplicate genes were removed to obtain the final set of tolDCs-RGs. The volcano maps of 4 groups of DEGs were plotted with 'ggplot2' (v 3.5.1) in R (v 3.5.1) [16] to visualize these DEGs, and the heatmaps of the expression of 4 groups of DEGs were plotted using the R package "ComplexHeatmap" (v 2.16.0) [17], with labeling and displaying top 10 up- and down-regulated genes sorted based on the differential ploidy $\log_2$FC.

## 2.3. Obtainment of candidate genes associated with tolDCs-RGs in GC

To attain the DEGs comparing GC samples and control samples in the training cohort, the R package "DESeq2" (v 3.56.2) [18] was used under the threshold of |log2FC|>2, P-value<0.01. The volcano of the top 10 DEGs was drawn via the R package "ggVolcano" (v 0.0.2) [19]. A heat map visualizing the top 10 differentially expressed genes ranked by $log_2FC$ from largest to smallest was drawn by using R packages "ComplexHeatmap" (v 2.16.0) [17]. The R package "ggvenn" (v 0.1.9) (https://CRAN.R-project.org/package=ggvenn) was employed to derive the intersection of the DEGs and the tolDCs-RGs. In this way, the candidate genes related to tolDCs-RGs in GC were identified.

## 2.4. Enrichment analysis of candidate genes

To systematically uncover the biological roles and pathway networks linked to candidate genes, we performed functional enrichment analyses through two complementary approaches. Pathway enrichment analysis based on the kyoto encyclopedia of genes and genomes (KEGG) database was performed to systematically detect metabolic and signaling pathways with significant enrichment. Gene Ontology (GO) functional annotation analysis was employed to systematically characterize the molecular roles of candidate genes. The GO analysis encompassed three primary categories: biological process (BP) to identify cellular and physiological processes, cellular component (CC) to determine subcellular localization and protein complexes, and molecular function (MF) to characterize biochemical activities and binding properties. These comprehensive functional analyses were undertaken through the R package "Clusterprofiler" (v 4.15.0.3) [20] with statistical significance determined at P<0.05.

To investigate protein-protein interactions and construct functional networks at the molecular level, candidate genes were systematically submitted to the Search Tool for the Retrieval of Interacting Genes (STRING) database (http://string-db.org), a comprehensive platform for protein association networks. The analysis was performed with a confidence level threshold of 0.4 to ensure reliable interaction predictions while maintaining network complexity. The resulting protein-protein interaction (PPI) network, including both direct physical interactions and functional associations, was subsequently imported and visualized using Cytoscape software (v 3.1.0) [21], allowing for detailed network topology analysis and identification of key hub proteins and functional modules within the interaction network.

## 2.5. Identification of prognostic genes

Multiple analytical approaches were implemented to screen prognostic genes with prognostic significance from candidate genes and build a risk model. First, the previously obtained candidate genes were subjected to univariate Cox regression analysis (P<0.05, Hazard Ratio (HR) ≠ 1) with proportional hazards (PH) assumption test (P>0.05) via "survival" package (v 3.7−0) [22]. The genes that met these criteria were recorded as potential prognostic genes. The results were visualized by drawing a forest plot with the R package 'forestplot' (v 3.1.5) [23]. Least absolute shrinkage and selection operator (LASSO) regression analysis was carried out for potential prognostic genes using the R package "glmnet" (v 4.3.2) in the training set to identify candidate prognostic genes [24] (family=cox, nlambda=1,000, alpha=1, nfolds=10). Genes were finally selected based on lambda. min and regression coefficients not equal to 0. Ultimately, a multivariate Cox proportional hazards model was developed via the coxph function from the 'survival' package (v 3.7−0), with the PH assumption validated by a statistical test (P>0.05). The model was optimized through stepwise regression analysis using the step function. The final model coefficients were used for prognostic genes with statistical significance.

## 2.6. Construction of risk model

The prognostic genes were employed to develop the risk prediction model for GC. The risk score was computed in the TCGA-STAD dataset using the formula:

$$\text{risk score} = \sum_{i=1}^{n} (\exp_i \times \beta_i)$$

[Note: For each prognostic gene, the transcriptome expression value is represented by $\exp_i$, while the coefficient in the stepwise analysis is indicated by $\beta$]. Based on comprehensive survival data and clinical follow-up information, GC patients from TCGA-STAD and validation datasets were systematically categorized into high-risk group (HRG) and low-risk group (LRG) according to the optimal risk score threshold determined through statistical analysis.

Survival analysis was performed through the Kaplan-Meier estimation approach implemented in the R package "survival" (v 3.7−0) for assessing survival disparities across different cohorts, employing log-rank testing to determine statistical significance (P<0.05). Risk score distribution curves and scatter plots were constructed to display the patterns of risk stratification and clinical outcome representation in each risk-defined patient subgroup.

To evaluate the discriminative performance of the prognostic model, time-dependent receiver operating characteristic (ROC) analysis was performed using the R package "timeROC" (v 0.4). This analysis covered clinically meaningful time points (1, 3, and 5 years). To ensure the reliability of the results, the bootstrap method was employed to calculate confidence intervals. The area under the curve (AUC) was computed to assess the predictive ability (AUC>0.6).

## 2.7. Construction of regulatory networks of GC

To explore the upstream regulatory mechanisms of the prognostic genes, we constructed both transcription factor-mRNA (TF-mRNA) and microRNA-mRNA (miRNA-mRNA) regulatory networks. Potential transcription factors (TFs) of prognostic genes were found by the Human Transcription Factor Targets database (htfTARGET, http://bioinfo.life.hust.edu.cn/hTFtarget#!/). After conditional filtering, TFs that appeared more than 5 times in the htfTARGET database were selected as the final results. A TF-mRNA molecular regulatory network was assembled utilizing Cytoscape software (v 3.1.0).

The mRNA-microRNA (miRNA) interactions of the prognostic genes were forecasted with the R package "multiMiR" (v 1.24.0) [25]. This R package contains 8 relevant databases: DIANA-microT-CDS (http://www.microrna.gr/microT), ElMMo, MicroCosm (https://www.ebi.ac.uk/enright-srv/microcosm/htdocs/targets/v5/), miRanda, miRDB (http://www.mirdb.org/), PicTar (http://pictar.mdc-berlin.de/), PITA (http://genie.weizmann.ac.il/pubs/mir07/mir07_data.html/), and TargetScan (http://www.targetscan.org/). Finally, the miRNAs predicted in four of the databases for the prognostic genes were selected as the final miRNAs, and the miRNA-mRNA regulatory network was assembled through the use of the Cytoscape software (v 3.1.0).

## 2.8. Clinical feature analysis

In the clinical characteristic analysis, to investigate the correlation between the risk score and various clinical parameters of gastric cancer patients, this study utilized data from GC patients with complete clinical information in the TCGA-STAD dataset. The risk scores were compared across subgroups of different clinical parameters. Specifically, age was stratified based on the median value (67 years) of the entire cohort into two groups:>67 years and ≥67 years, to ensure balanced sample sizes between groups and enhance the statistical power. Other clinical features were categorized according to their standard classification systems, including gender (male, female), pathological T stage (T1, T2, T3, T4), N stage (N0, N1, N2, N3), and M stage (M0, M1). The Wilcoxon test (P<0.05) was applied to compare risk score variations among subgroups with distinct clinical characteristics, thereby investigating risk score associations with GC patient clinical features. And the box plot was drawn employing the R package "ggplot2" (v 3.5.1) to illustrate the results.

## 2.9. Immune infiltration analysis

First, to examine the infiltration levels of immune cells in the HRG and LRG, the relative abundance of 28 immune cells [26] in both groups was assessed using the R package "GSVA" (v 1.53) [27]. Then, the Wilcoxon test was conducted with

the R package "rstatix" (v 0.7.2) [28] to detect divergences in immune cell abundance when comparing HRG with LRG, and to identify significantly different immune cells (P<0.05). Subsequently, Spearman's rank correlation analysis was performed on the differentially expressed immune cells using the R package "psych" (v2.1.6) [29]. The p-values were adjusted using the Benjamini-Hochberg method, with significance thresholds set at a correlation coefficient absolute value |cor|>0.30 and an adjusted P-value<0.05. Then, integrating the prognostic gene set and differentially abundant immune cell populations from TCGA-STAD, the relationship between them was evaluated using the Spearman analysis, and heatmaps were constructed by means of the R package 'ggplot2' (v 3.5.1) (|cor|>0.30, adj.P<0.05).

### 2.10. Gene set enrichment analysis (GSEA)

To identify the underlying biological functions and pathways that differed between the high-risk and low-risk groups, we performed Gene Set Enrichment Analysis (GSEA) on the samples from both groups. Comparative analysis of HRG and LRG in TCGA-STAD was analyzed using the R package "DESeq2" (v 3.56.2), a robust statistical framework specifically designed for RNA-sequencing count data normalization and differential expression detection. Following differential expression profiling, the resulting gene list was systematically arranged in descending order based on log2 fold change values to establish a pre-ranked gene signature for subsequent pathway enrichment analysis. The genes were sorted (ranked by $log_2FC$ from high to low) and then enriched using the GSEA function in the R package "clusterProfiler" (v 4.15.0.3) [20]. which enables comprehensive functional annotation and pathway enrichment assessment. The reference gene set collection, designated as "c2.cp.kegg.v7.4.symbols.gmt", was obtained from the Molecular Signatures Database (MSigDB) portal (https://www.gsea-msigdb.org/gsea/index.jsp). Stringent selection criteria were applied to identify statistically significant enriched pathways, including adj.P<0.05 for multiple testing correction and Normalized Enrichment Score (|NES|) > 0 to ensure biological relevance. The most significantly enriched pathways (top 5) were visualized and presented through sophisticated graphical representations generated using the R package "ggplot2" (v 3.5.1), providing clear visual interpretation of the functional enrichment results.

### 2.11. Drug sensitivity analysis

To probe the therapeutic effects of drugs on patients in HRG and LRG, 297 commonly used chemotherapeutic agents for gastric cancer were retrieved from the Genomics of Drug Sensitivity in Cancer (GDSC) database (https://www.cancerrxgene.org). In the TCGA-STAD dataset, by utilizing the R package "pRRophetic" (v 0.5) [30], the IC50-based drug response predictions for the standard chemotherapeutic agents in HRG and LRG were computed. Furthermore, IC50 differences across risk differences across groups were assessed via the Wilcoxon rank-sum test (P<0.05). Generally, the IC50 value was negatively correlated with drug sensitivity. Specifically, the lower the IC50 value, the stronger the inhibitory effect of the drug can be exerted at a lower concentration, and the higher the drug sensitivity; conversely, the higher the IC50 value, the lower the drug sensitivity. Spearman's correlation was employed to evaluate the association between the risk score and drug sensitivity, with Benjamini-Hochberg correction applied (|correlation (cor)|>0.30, adj.P<0.05).

### 2.12. Expression analysis

To examine prognostic gene expression patterns in GC patients versus normal controls, the Wilcoxon rank test (P<0.05) was applied to determine prognostic gene expression disparities in GC compared to normal groups in the training datasets, with box plots generated via "ggplot2" (3.5.1) for visualization.

### 2.13. Single-cell RNA sequencing data characterized the gene expression patterns of cells within the gastric cancer microenvironment

To characterize gene expression patterns within the gastric cancer microenvironment at single-cell resolution, a systematic analysis was performed using the GSE183904 dataset from the GEO database and the Seurat software package

(v5.1.0) [31]. Data quality control was first conducted by calculating metrics including gene counts, cell counts, and the percentage of mitochondrial genes. Genes covered by fewer than 30,000 cells were filtered out, and high-quality cells retaining between 600 and 4,000 expressed genes with less than 20% mitochondrial gene content were preserved. Violin plots generated by the VlnPlot function were used to visualize the quality metrics before and after filtering.

During the dimensionality reduction and clustering phase, data integration was performed using the IntegrateData function to mitigate batch effects. Normalization was carried out via the NormalizeData function with a scale factor of 10,000. The FindVariableFeatures function with the "vst" method was then employed to identify the top 2,000 highly variable genes (HVGs) for downstream analysis. Principal component analysis (PCA) was conducted using the RunPCA function based on these HVGs. The Jackstraw function with 100 permutations (P<0.05) was applied alongside the ElbowPlot to determine the optimal number of principal components. Subsequently, a k-nearest neighbor graph was constructed using the FindNeighbors function, and cell clustering was performed with the FindClusters function at a resolution of 0.4. Finally, Uniform Manifold Approximation and Projection (UMAP) was used for nonlinear dimensionality reduction, and cell distribution patterns were visualized with the DimPlot function.

For cell type annotation, the FindAllMarkers function was first utilized to identify cluster-specific differentially expressed genes (thresholds: logfc.threshold=0.25, min.pct=0.25, only.pos=TRUE). Marker genes from the literature [32] were also referenced. Cell subpopulations were systematically annotated using the SingleR package (v2.4.0) and the CellMarker 2.0 database. Dot plots were generated to display marker gene expression patterns, UMAP plots were used to present the annotation results, and bar plots were created to compare the distribution of different cell types between the normal and gastric cancer groups.

Finally, based on the cell annotation results, the expression characteristics of prognosis-associated genes across various cell types were analyzed. Wilcoxon tests were employed to compare the expression differences of key genes between the normal and gastric cancer groups, and box plots were generated for visualization. Particular attention was paid to the expression patterns of key genes in dendritic cells, providing single-cell-level evidence for a deeper understanding of the biological functions of these genes within the gastric cancer microenvironment.

## 2.14. Statistical analysis

All statistical analyses were performed in the R programming environment (v 4.2.2), which provides comprehensive support for data processing, statistical modeling, and result visualization. The specific analytical methods and corresponding tools were as follows: differential expression analysis was conducted using the "limma" package (v 3.56); gene function and pathway enrichment analyses were performed with "ClusterProfiler" (v 4.15.0.3); univariate and multivariate Cox regression analyses were carried out via the "survival" package (v 3.7−0); group comparisons were assessed using the Wilcoxon rank-sum test, implemented with the "rstatix" package (v 0.7.2); correlations between variables were evaluated by Spearman's rank correlation analysis using the "psych" package (v 2.1.6); in drug sensitivity analysis, the "pRRophetic" package (v 0.5) was utilized to predict the IC50 of chemotherapeutic agents. Unless otherwise specified, all hypothesis tests were two-sided, and statistical significance was defined as P<0.05.

## 2.15. Ethics approval and consent to participate

Not applicable.

## 3. Results

### 3.1. Identification of tolDCs-RGs

To obtain a comprehensive set of tolerogenic dendritic cell-related genes, differential expression analysis was performed across four independent datasets. The results showed that 553 (179 down-regulated and 374 up-regulated), 607 (260

down-regulated and 347 up-regulated), and 15 (1 down-regulated and 14 up-regulated) DEGs were identified in tolDCs-DEGs1, tolDCs-DEGs2, and tolDCs-DEGs3, respectively (Figs 1A and S1A-B), while no significant DEGs were detected in tolDCs-DEGs4 (S1C Fig). By integrating the DEGs from the first three datasets and removing redundancies, a final set of 1003 tolDCs-RGs (S1 Table) was established, providing a critical foundation for subsequent analyses.

### 3.2. Identification and enrichment analysis of candidate genes

To identify candidate genes associated with the function of tolerogenic dendritic cells in gastric cancer, 1,464 differentially expressed genes (DEGs) were screened between the GC and control groups. Among these, 680 genes were up-regulated and 784 were down-regulated in the GC group (Fig 1B–1C). The intersection of the above DEGs and tolDCs-RGs was taken to obtain the 46 candidate genes related to tolDCs in GC (Fig 2D). Further functional enrichment analysis of these candidate genes revealed that the GO analysis yielded 510 biological processes, 20 cellular components, and 67 molecular functions that were significantly enriched ($P < 0.05$). (S2 Table), of which the top 5 most significant pathways (ranked by P value from lowest to highest) in each section were displayed. Specifically, for BP, the top pathways included chemokine-mediated signaling pathway, response to chemokine, etc; for CC, included specific granule, specific granule membrane, cluster of actin-based cell projections, and so on; for MF, included CXCR chemokine receptor binding, chemokine activity, and so on. (Fig 2E). Furthermore, the candidate genes confirmed substantial enrichment in 18 KEGG pathways ($P < 0.05$) (S3 Table). The top results included IL-17 signaling pathway, Toll-like receptor signaling pathway, and cytokine-cytokine receptor interaction, among others (Fig 2F). Subsequently, we investigated the protein-protein interactions. After excluding 12 unconnected target genes, the PPI network results showed that the proteins encoded by the 34 remaining genes shared 116 connecting edges. Among these, CXCL8 exhibited the most prevalent associations at the protein level with other genes like SCIN and GDF15 (Fig 2G). These results suggested that the identified candidate genes might be linked to immune function, thereby providing clues for further exploration of the mechanisms by which tolerogenic dendritic cells function in the gastric cancer microenvironment.

### 3.3. Screening of prognostic genes

To construct a prognostic risk assessment model for gastric cancer, we performed a multi-stage screening process based on 46 candidate genes. First, 12 potential prognostic genes significantly associated with survival were identified through univariable Cox regression (HR ≠ 1 and $P < 0.05$) in combination with the proportional hazards assumption test ($P > 0.05$) (Fig 2A). The 8 LASSO genes were obtained by LASSO (RNASE1, PDK4, CXCL1, GPX3, INHBA, NFE2L3, ASCL2, CD36) (Fig 2B) at log(lambda.min) = 0.019. The 5 prognostic genes (RNASE1, CXCL1, GPX3, INHBA, and ASCL2) were finally obtained by multivariate stepwise regression (Fig 2C). Based on the expression levels of these genes and their regression coefficients, the following risk score formula was established: Risk Score = (0.11550940) × RNASE1 + (−0.9954886) × CXCL1 + (0.15820155) × GPX3 + (0.12561823) × INHBA + (0.08923591) × ASCL2. This laid the foundation for the subsequent construction of the risk model.

### 3.4. Construction of risk assessment

Based on the determined risk score, we first stratified the gastric cancer patients in the TCGA-STAD training set. Using the optimal cut-off value of the risk score (0.9974999) as the threshold, the GC patient samples in the TCGA-STAD training set were classified into a high-risk group (n = 198) and a low-risk group (n = 163). Fig 3A demonstrated the survival times and risk scores for each sample in the HRG and LRG; the results showed an obvious excess of deaths in the HRG over the LRG. The K-M survival curves demonstrated that individuals in the HRG had significantly poorer survival compared to those in the LRG (Fig 3B). This strongly suggested that there was a strong link between risk grouping and survival outcomes. The ROC curves for survival at 1, 3, and 5 years showed that AUC values were > 0.6 in 1, 3, and 5

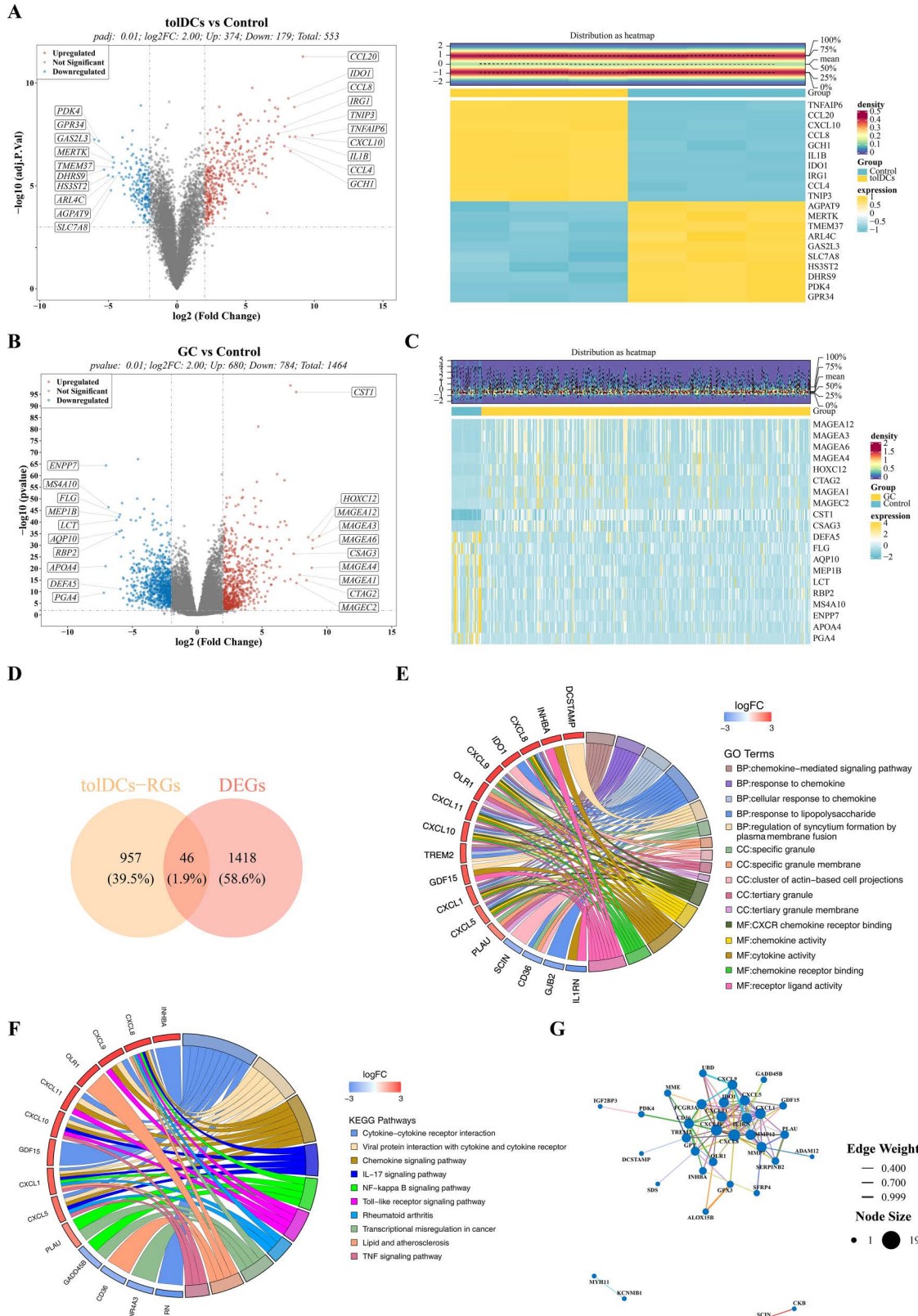

**Fig 1. Identification and enrichment analysis of candidate genes.** (A) Volcano map and expression heat map of differentially expressed genes of ToldC-DEGS1. (B) DEGs distribution volcano maps of GC. (C) DEGs expression heat maps of GC. (D) 46 candidate genes associated with tolDCs in

GC were identified. (E) Association of candidate genes with biological processes, cell components, and molecular function. (F) The top 10 KEGG pathways with significant enrichment of candidate genes. (G) The network interaction of candidate genes at the protein level.

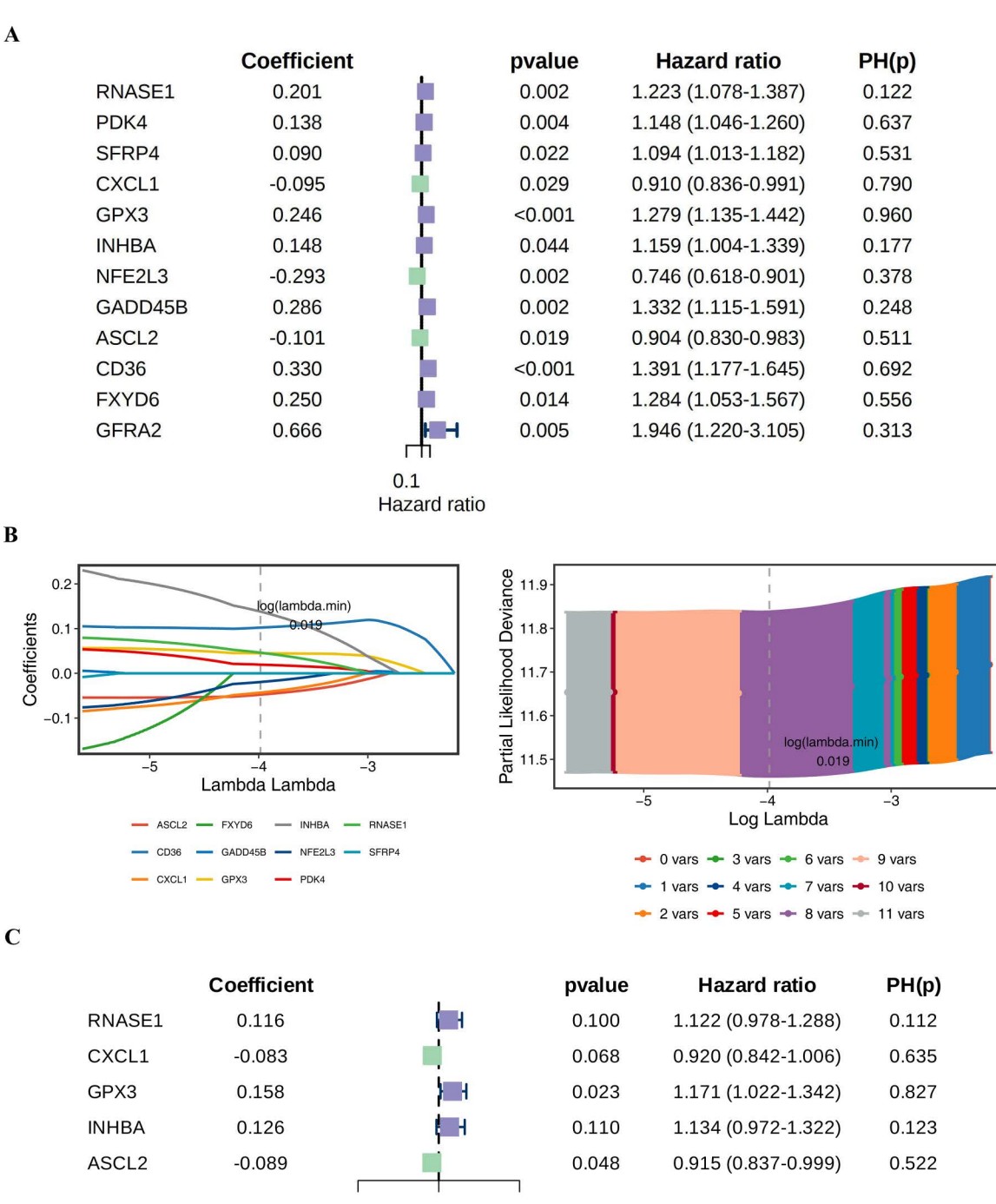

**Fig 2. Potential prognostic genes were determined by univariate Cox regression analysis, LASSO regression analysis, and stepwise regression analysis.** (A) The forest map of 12 potential prognostic genes obtained by univariate Cox regression analysis. (B) LASSO regression analysis was performed on 12 potential prognostic genes. (C) The forest map of 5 prognostic genes obtained by stepwise regression analysis was finally obtained.

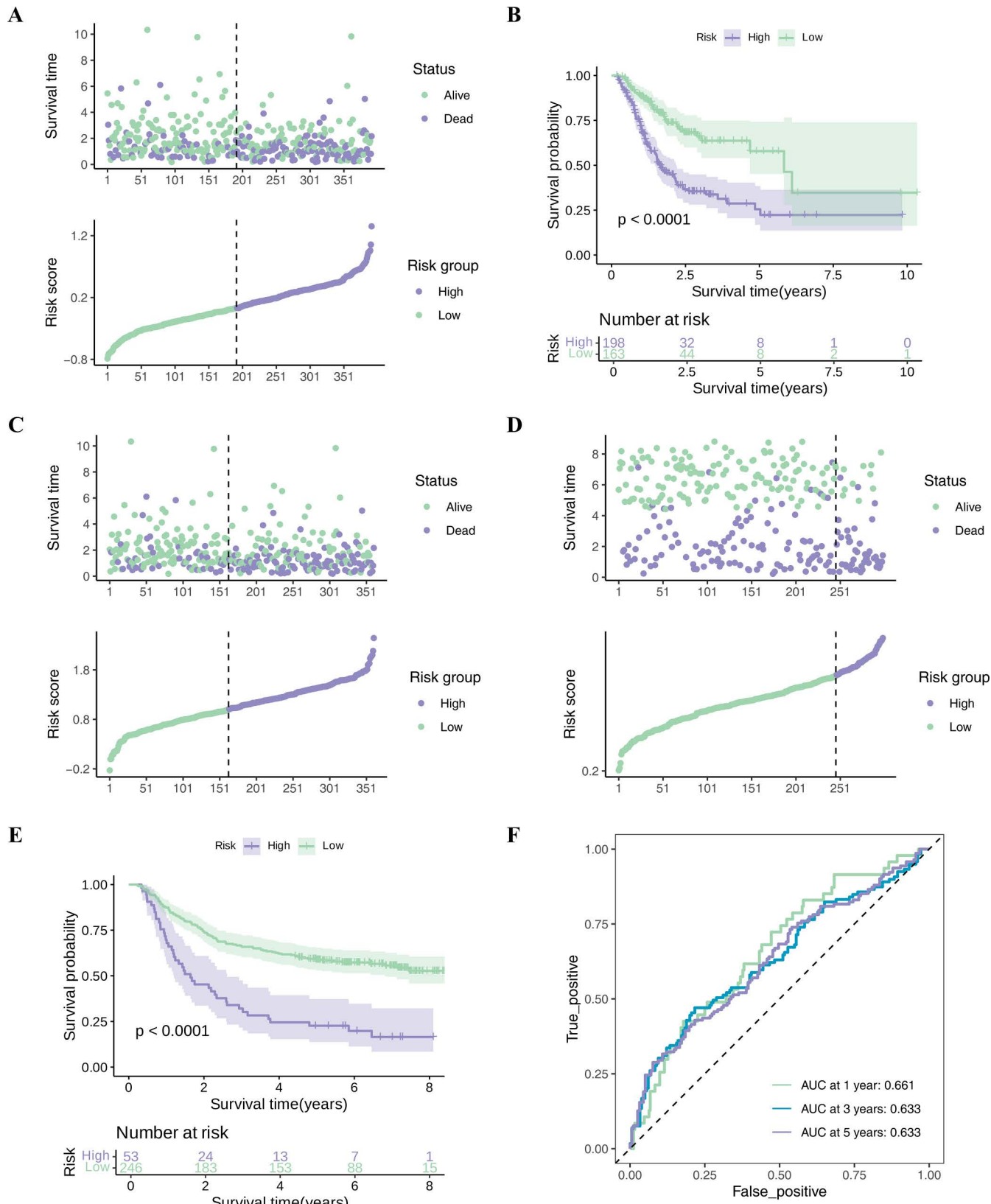

**Fig 3. The construction and evaluation of risk assessment.** (A) Survival status of GC patients in high and low risk groups in TCGA-STAD training set. (B) Survival curve of GC patients in high and low risk groups in TCGA-STAD training set. (C) ROC curve of 1-, 3-, and 5-year survival rates in the

TCGA-STAD training set. (D) The survival status of GC patients in the high and low risk groups in the validation set. (E) Survival curve of GC patients in high and low risk groups in the validation set. (F) ROC curve of 1-, 3-, and 5-year survival rates in the validation set.

years, suggesting that the model's ability to discriminate between survival statuses at the 1, 3, and 5-year time points was excellent (Fig 3C).

To evaluate the model's robustness, we conducted an identical analysis on the independent validation set GSE66229. Each patient in the validation set received a risk score calculated through the same formula, leading to the classification of GC patients into HRG (53) and LRG (246) using the best cutoff value (0.6149928). There was a statistically significant disparities between HRG and LRG for survival status in GC patients (Fig 3D). The K-M survival curves in the validation set had the same effect as the training set; the LRG maintained a relatively high survival probability over a longer period (P<0.0001) (Fig 3E). Finally, the ROC curves for the OS of GC patients at 1, 3, and 5 years, and the AUC values were all above 0.6, suggesting that the prediction accuracy was satisfactory (Fig 3F). These results collectively demonstrate that the five-gene-based risk model could effectively distinguish the prognostic risk of gastric cancer patients to a certain extent, and possessed a degree of reliability and generalizability.

### 3.5. Exploring the potential regulatory mechanisms of GC

We analyzed the potential transcriptional factors (TFs) and miRNA regulatory network using the hTFtarget database. The results identified 18 TFs that may regulate these genes, such as KDM5B, FOXA2, and GATA3. Among them, 5 TFs corresponded to RNASE1, 7 to ASCL2, 12 to INHBA, 7 to CXCL1, and 9 to GPX3 (Fig 4A). At the post-transcriptional level, we further predicted miRNAs targeting these genes. The results indicated that 86 miRNAs might be involved in the regulation of these prognostic genes. The distribution included 1 miRNA for ASCL2, 15 for GPX3, 23 for CXCL1, 2 for RNASE1, and 45 for INHBA (Fig 4B). Examples include hsa-miR-1233-3p for INHBA and hsa-miR-582-5p for CXCL1. These findings shed initial light on the multi-level regulation that these five prognostic genes may be subjected to, providing direction for a deeper understanding of their molecular mechanisms in gastric cancer progression.

### 3.6. Clinical feature

The P-value of 0.018 (P<0.05) for age group less than 67 years and age group more than 67 years indicated that risk scores differed significantly across age groups, while the risk score value for gender was 0.56, suggested that there was no significant disparity between genders (male and female) (Fig 4C-4D).

Moreover, significant risk score disparities were observed between some of the levels in the M, N, and T stages (P<0.05). The risk score value of 0.013 between M0 and M1 demonstrated significant differences between these groups, similarly, there was a notable disparity among N1 and N2 (P=0.024), T1 and T2 (P=0.00088), T3 (P=0.0042), T4 (P=$6.4 \times 10^{-5}$) (Fig 4E-4G). This suggested that the clinical stage was potentially important in assessing the risk and could assist in determining the degree of risk of the patient.

### 3.7. Exploring immune infiltration differences

The distribution of immune cells was illustrated through heatmap visualization (Fig 5A). Employing the Wilcoxon test for statistical analysis, 20 significantly different immune cells were identified with statistical significance set at adj.P<0.05. In particular, activated B cells in HRG demonstrated substantially elevated infiltration levels (P<0.0001), while activated CD4 T cells in LRG showed markedly enhanced infiltration (P<0.01). According to the box plot analysis results, there were notable marked variations in the tumor-infiltrating immune cell patterns between the two groups. The box-and-whisker plot visualization indicated that the degree of cell infiltration was predominantly higher in the HRG compared to the LRG (Fig 5B). Via correlation analysis methodology, the majority of differential immune cells showed statistically significant

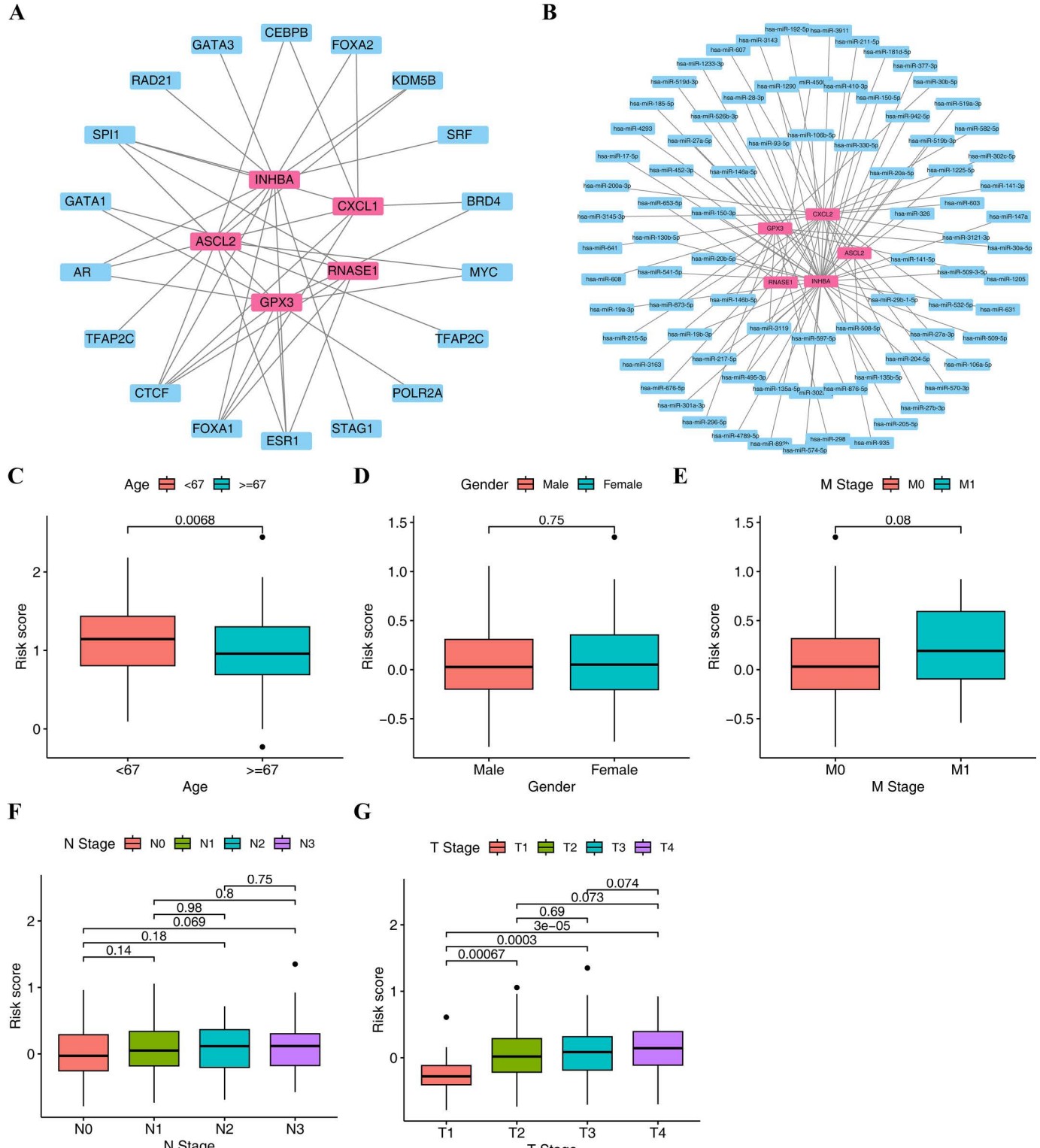

**Fig 4. The potential regulatory mechanisms of GC and the correlation of risk score in different clinical features.** (A) TF-mRNA network. (B) miRNA-mRNA network. (C) Risk score and age difference analysis. (D) Risk score and gender difference analysis. (E) Analysis of the difference between risk score and M stage. (F) Analysis of the difference between risk score and N stage. (G) Analysis of the difference between risk score and T stage.

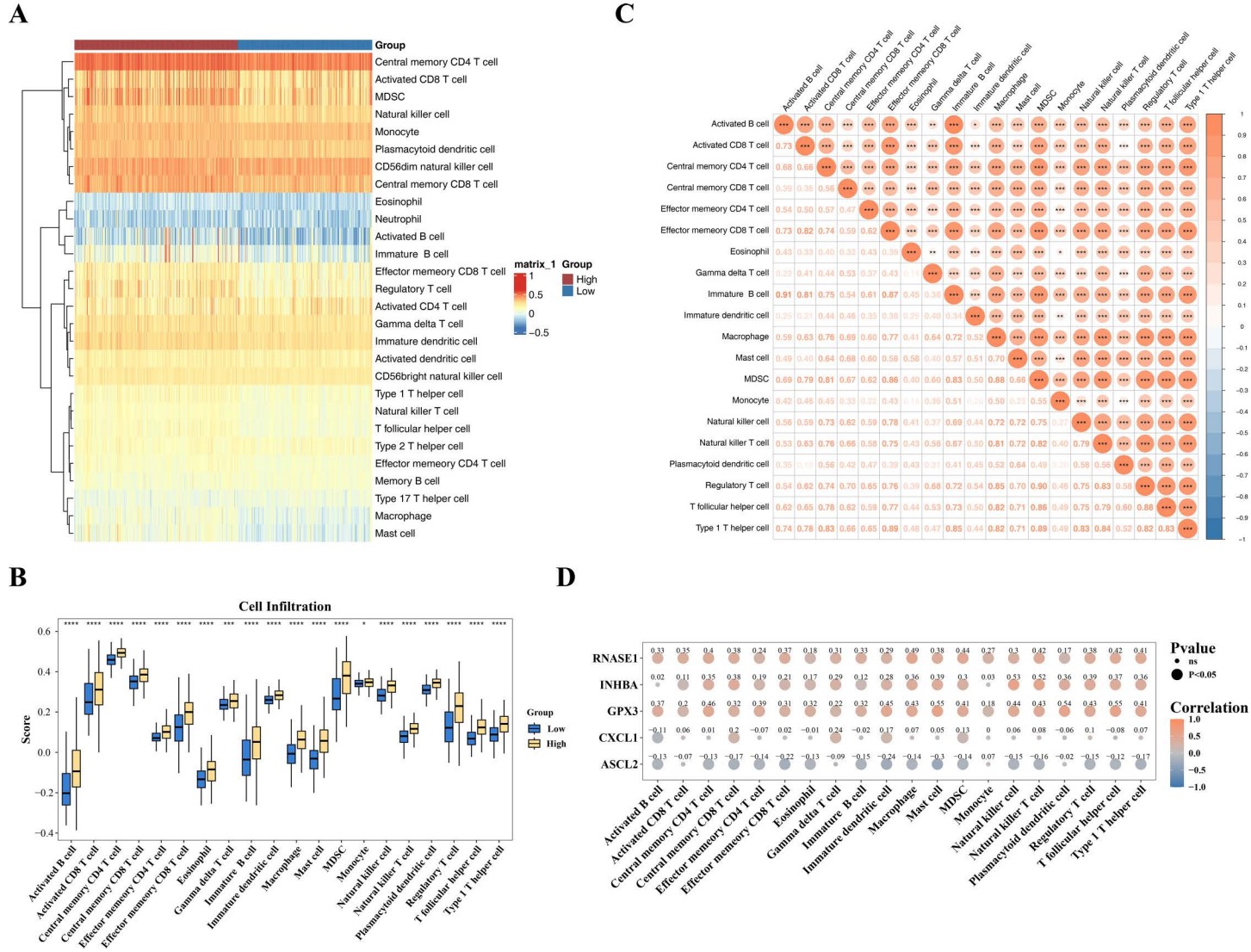

**Fig 5. Difference of immune infiltration.** (A) Heat map of immune cell infiltration in high and low risk groups of patients. (B) Difference analysis of immune cells in GC high and low risk group samples. (C) Correlation analysis between different immune cells. (D) Correlation analysis between differential immune cells and prognostic genes.

positive associations (cor > 0.3, with significance threshold at P < 0.05). As an illustrative example, immature B cells and activated B cells exhibited a strong positive correlation (cor = 0.92, achieving statistical significance at P < 0.001) (Fig 5C). These results revealed a complex interrelationship among differential immune cells, which might reflect changes in the GC immune microenvironment.

The association of prognostic genes with differential immune cells in the training set was evaluated by Spearman correlation analysis, with bubble plots illustrating the correlation coefficients and significance levels. The findings indicated that GPX3 demonstrated a direct positive relationship with the majority of cells, encompassing regulatory T cells. ASCL2 was weakly correlated with most immune cells (Fig 5D). These results suggested a complex correlation between prognostic genes and immune cells, which might reveal their potential interactions in the GC microenvironment.

### 3.8. GSEA result

To explore the biological characteristics of the different risk groups, we performed GSEA to examine the pathway enrichment between the high-risk and low-risk groups.GSEA showed that HRG was enriched in a total of 30 pathways. Five pathways were significantly enriched, including hypertrophic cardiomyopathy and dilated cardiomyopathy, among others (Fig 6A) (S4 Table). LRG was enriched in a total of 17 pathways. The top 5 significantly enriched pathways were the peroxisome pathway, pyrimidine metabolism pathway, RNA polymerase pathway, and steroid hormone synthesis pathway (Fig 6B) (S4 Table).The divergent enrichment of pathways between the two groups suggested distinct biological characteristics in high-risk versus low-risk gastric cancer patients: the high-risk group was associated with certain cardiovascular disease pathways, whereas the low-risk group exhibited more active metabolic and biosynthetic processes. These findings provide new clues for understanding the molecular mechanisms underlying the different risk subgroups.

### 3.9. Exploration of the effectiveness of drug therapy

Drug sensitivity analysis results showed that the difference in IC50 between HRG and LRG for each drug; the lower the IC50, the greater the sensitivity to the patient. The top 5 drugs with down-regulation of IC50 and the top 5 drugs with up-regulation of IC50 were collated separately (P<0.05). BIBW2992, GW.441756, and VX.680, etc., 5 drugs had higher IC50 in the HRG than in the LRG, and there were significant differences between all 5 drugs in both the HRG and LRG (P<0.0001). While dasatinib, DMOG, etc., 5 drugs had higher IC50 in the LRG than in the HRG (P<0.0001) (Fig 6C). BIBW2992 displayed a significant positive association with risk score (cor>0.3, P<0.05) by Spearman correlation analysis (Fig 6D). These findings suggested that the classification based on the risk model could provide a reference for drug selection in gastric cancer patients, and that patient groups with distinct risk profiles might be suited for differential treatment strategies.

### 3.10. Single-cell RNA sequencing data were analyzed and expression differences of key genes across different cell types were identified

Single-cell RNA sequencing analysis of the GSE183904 dataset yielded 134,563 high-quality cells after quality control (retention rate 84.8%) (S2A-B Fig), with the gene expression matrix remaining intact. Through identification of 2,000 highly variable genes and UMAP dimensionality reduction clustering (S2C-F Fig), cells were successfully partitioned into 22 distinct subpopulations, which were further annotated into 11 cell types including T cells, dendritic cells, and epithelial cells (S2G-I Fig). Cell proportion analysis showed epithelial cells as the most abundant population, while dendritic cells showed an increased proportion in the gastric cancer group, suggesting their potential involvement in tumor microenvironment regulation (S2J Fig).

Analysis of the key gene expression patterns (Fig 7) showed that the expression levels of INHBA, ASCL2, and CD36 in dendritic cells of the gastric cancer group were significantly upregulated. Such significant changes in expression indicate that these genes may play important roles in the regulation of dendritic cell function. In addition, ASCL2 and CD36 were specifically overexpressed in chief cells, while ASCL2 and INHBA were significantly upregulated in neutrophils. These results suggest that although the prognostic signature we constructed was derived from the screening of tolDC - related genes, its role in the gastric cancer microenvironment may not be limited to dendritic cells, but rather is achieved through the coordinated interaction of multiple cell types. These findings validate the expression specificity of prognostic genes at the single – cell resolution, especially suggesting the potential regulatory functions of INHBA, ASCL2, and CD36 in gastric cancer, thus providing a new perspective for understanding how these genes affect gastric cancer progression through the regulation of the immune microenvironment.

## 4. Discussion

TME encompasses non-cancerous cellular constituents and their secreted molecules, collectively shaping the local TME. Extensive immune cell infiltration is observed in the tumor region, such as tolDCs. These immune cells constitute the

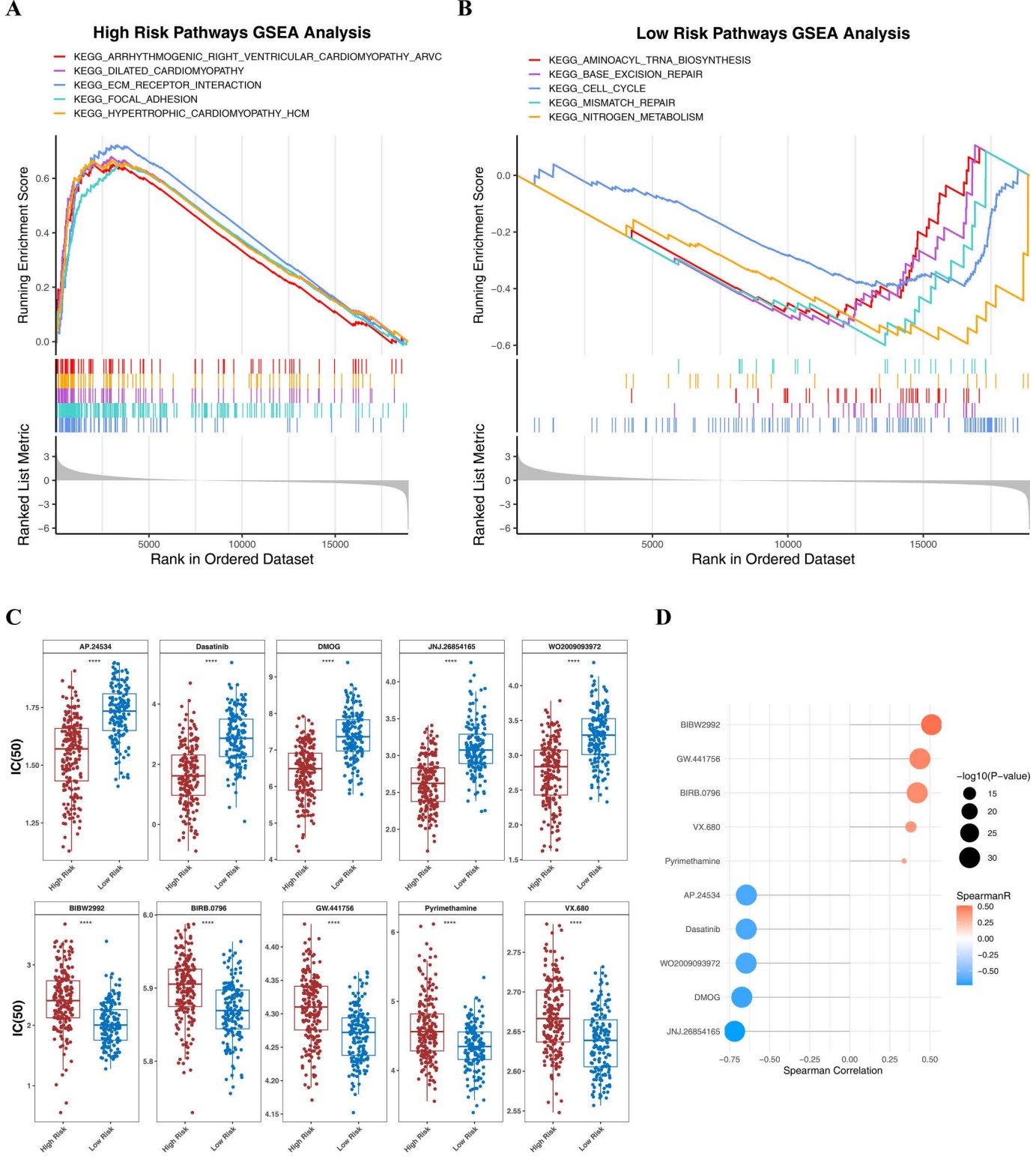

**Fig 6. GSEA enrichment analysis and drug sensitivity analysis.** (A) The top five most enriched KEGG pathways in high-risk group. (B) The top five most enriched KEGG pathways in low-risk group. (C) The difference between the top 5 drugs with IC50 up-regulation in the high (red) and low (blue) risk group. The difference between the top 5 drugs with IC50 down-regulation in the high and low risk group. (D) Correlation between risk score and drugs.

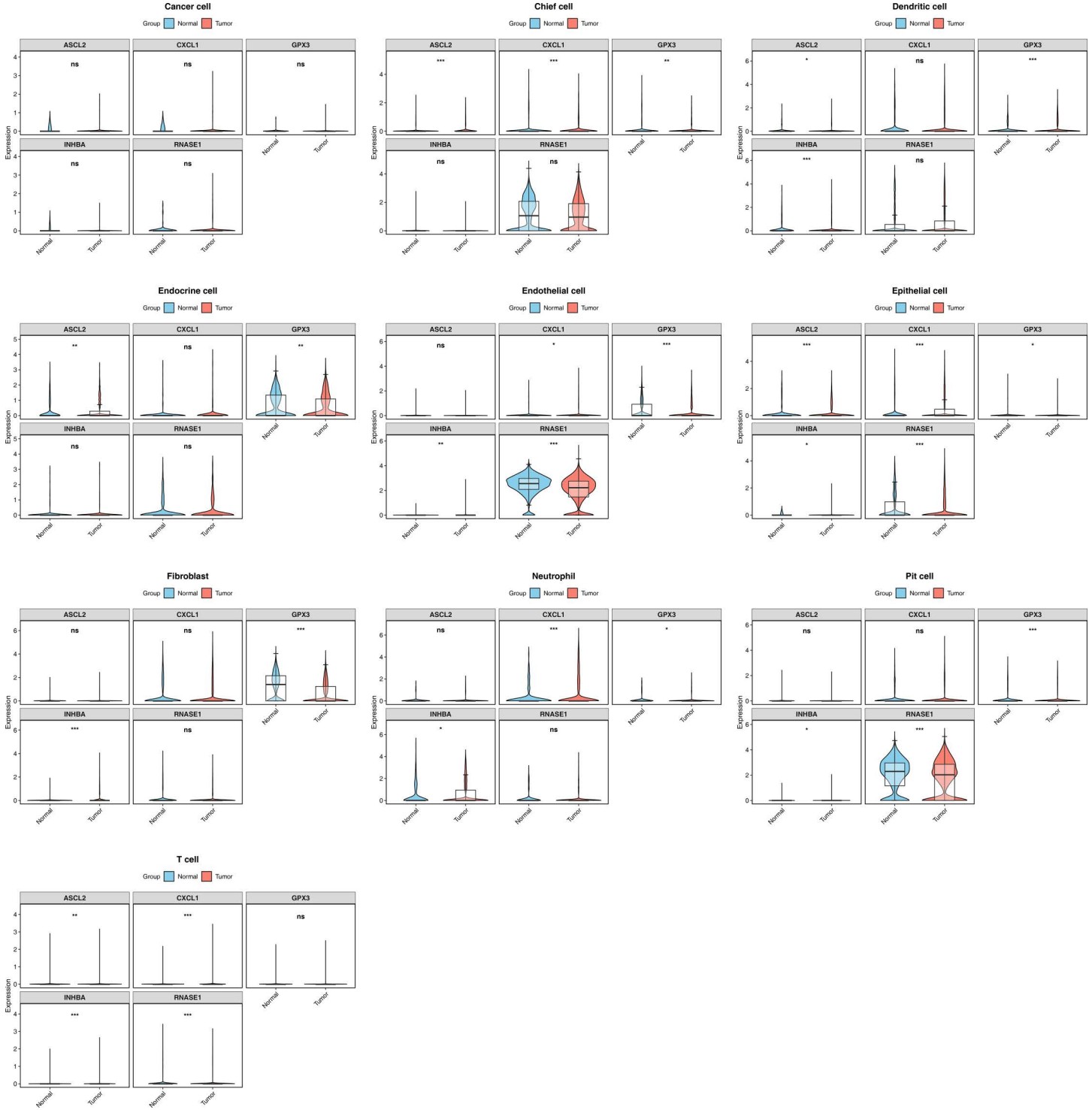

**Fig 7. Expression differences of key genes across different cell types.** The blue color represents the normal control group, and the red color represents the gastric cancer group. ns, not significant; *, p<0.05; **, p<0.01; ***, p<0.001.

tumor immune microenvironment. Current studies have confirmed that persistent interactions between neoplastic cells and the surrounding tumor microenvironment play a pivotal role in the pathogenesis, evolution, metastatic dissemination, and therapeutic sensitivity of malignancies [33]. Long-term high gastric acid environment may constitute a unique immune microenvironment in the stomach. As a special kind of DC, tolDC plays an important role in immune tolerance and immunosuppression, it has been reported that the tolDCs contributes to the immune escape of cancer cells from the host anti-tumor response in various malignant tumors [34,35]. The increase of peripheral tolDCs in patients with gastric cancer may contribute significantly the immunosuppression of patients with GC [36]. Therefore, based on the use of systematic bioinformatics methods, we first screened tolDCs-DEGs by differential expression analysis, and then performed differential analysis through the GC and Control group of the training set, and intersected the analysis results with tolDCs-DEGs to obtain 46 candidate genes. Subsequently, in order to further analyze the fuctional we constructed a PPI network for visual analysis, and verified its biological function through gene enrichment analysis. In terms of prognostic analysis, we applied univariate Cox regression analysis to identify potential prognostic related genes, and then combined with stepwise regression algorithm for further screening. Finally, we successfully identified five key genes with prognostic value (CXCL1, INHBA, ASCL2, RNASE1 and GPX3), and developed a prognostic risk prediction model for gastric cancer based on these genes.

C-X-C motif ligand 1 (CXCL1) chemokine, alternatively referred to as growth-regulated gene alpha (GRO-α), is one of the CXC chemokine family. CXCL1 gene is a protein-coding gene which located on 4q13.3 and clustered with other CXC chemokine genes. CXCL1 is mainly expressed by macrophages, neutrophils, and epithelial cells, and exerts biological functions via binding to chemokine receptor C-X-C motif receptor 2 (CXCR2). Activation of CXCR2 by CXCL1 can induce cell migration and trigger signal transduction [37]. CXCL1 may be produced by tumor cells and is involved in the occurrence, tumor angiogenesis, and cell migration of a variety of malignant tumors, including head and neck cancer, liver cancer, and colon cancer. [38–40]. CXCL1 has been shown to induce the expansion of polymorphonuclear myeloid-derived suppressor cells (PMN-MDSCs) in GC, and has revealed the mechanism by which PMN-MDSCs promote the depletion of CD8＋T cells, a subset of T lymphocytes, which may be associated with the resistance of gastric cancer patients to immune checkpoint inhibitors (ICIs) [41]. The expression of CXCL1 can promote the mobilization of human GC-originated mesenchymal stem cells (hGC-MSCs), thereby participating in the remodeling of the tumor microenvironment and promoting the advancement and metastatic spread of GC through immunosuppression [42]. In addition, the activation of CXCL1-CXCR2 signaling axis showed a significant positive correlation with the malignant progression of gastric cancer. Clinical observation also found that CXCL1 showed high frequency expression in gastric cancer tissues. This phenomenon further supports that CXCL1 may play a key driving role in the invasion, metastasis and malignant progression of gastric cancer by activating CXCR2-mediated downstream signaling pathways [43].

As a multifunctional cytokine, Inhibin Subunit Beta A (INHBA) belongs to the transforming growth factor-beta (TGF-β) superfamily, which modulates a range of cellular processes, including proliferation, differentiation, adhesion, and migration across various cell types through an intricate signaling network [44]. INHBA plays a critical role in various cancers [45,46]. INHBA gene silencing can inhibit the TGF-β pathway and reduce the phosphorylation level of key proteins, thereby inhibiting the growth, motility and invasiveness of GC cells [47]. Circular RNA circTHBS1 can drive GC progression by up-regulating INHBA expression in a ceRNA and RBP-dependent manner and activating TGF-β signaling pathway [48]. On the other hand, the βA subunit encoded by the two INHBB genes can also form a subtype of activin. A cohort studies have shown that activin tumor protein expression is higher and overall survival is longer in the Adenocarcinomas of the esophagus (AEG) and stomach (AS) tumor patient cohort [49].

Achaete-scute family bHLH transcription factor 2 (ASCL2), a member of the basic helix-loop-helix (bHLH) transcription factors, plays a critical role in the maintenance of adult intestinal stem cell identity as a downstream effector of the Wnt signaling pathway [50]. Overexpression of ASCL2 correlated with an unfavorable clinical outcome and drug resistance in a variety of tumors [51–53]. Clinical data show that high expression of ASCL2 is closely correlated with the poor prognosis

 

of patients with GC, especially in patients with advanced (III + IV) GC, a significantly decreased survival rate was observed in the ASCL2 high-expression group when compared to the low-expression group [54]. Functional experiments further confirmed the key role of ASCL2 in the malignant progression ofGC. After knocking down the expression of ASCL2 by lentiviral vector, the migratory and invasive capacities of GC cells were markedly diminished, and the colony formation ability was also significantly inhibited. In vivo experiments revealed that knockdown of ASCL2 could effectively suppress the tumorigenic ability of GC cells in nude mice, and both tumor size and mass were markedly decreased [55]. Mechanism studies have revealed that ASCL2, as an epigenetic switch driven by Wnt signaling, maintains the characteristics of gastric cancer stem cells (CSCs) through SMYD3-mediated H3K4me3 modification, thereby promoting tumor growth and metastasis [56]. Therefore, ASCL2 is not only an important biomarker for the prognosis evaluation of gastric cancer, but also a potential target for precise treatment of gastric cancer..

Glutathione Peroxidase 3 (GPX3), a known glycosylated selenocysteine protein, is an important member of the glutathione peroxidase family. By reduced glutathione, GPX3 can catalyze the reduction of hydroperoxides, including $H_2O_2$ and soluble lipid hydroperoxides, and convert them into harmless water or alcohols, thereby protecting cells from oxidative stress damage [57]. GPX3 is the only GPx found in plasma. GPX3 is widely expressed in various tissues, especially in the kidney, heart, lung, and liver, and can be secreted into plasma to participate in extracellular antioxidant defense. We found that GPX3 was positively correlated with the individual risk ofGC. Studies have found that GPX3 contribute to tumor progression through the promotion of immune cell infiltration and the activation of immune checkpoint pathways [58]. Such a conclusion is supported by the data obtained in this study. Other studies have shown that GPX3 prevents GC cells migration and invasion by targeting the NF-κB/Wnt5a/JNK signaling pathway [59].

Ribonuclease A Family Member (RNASE2) belongs to the RNaseA superfamily, alternatively named Eosinophil-Derived Neurotoxin (EDN). It is among the four principal secretory proteins secreted by eosinophils after activation. Its protein products play a direct antibacterial role by degrading the RNA of pathogens, and can be used as chemokines to recruit other immune cells to the site of infection or inflammation. [60,61]. RNASE2 could result in maturation, activation and directional migration of DC and induce the expression of multiple pro-inflammatory cytokines and chemokines [62,63]. RNASE2 mRNA expression is significantly elevated in PBMCs from patients with systemic lupus erythematosus (SLE), and correlates with disease activity and autoantibody levels. It can also promote the production of age-associated B cells (ABCs) [64]. RNASE2 can also be used as a molecular marker to evaluate eosinophil-mediated airway inflammation and microvascular leakage [65]. Several bioinformatics studies have shown that RNASE2 is significantly associated with the prognosis of gastric cancer by affecting the tumor microenvironment [66,67]. These findings align with the results of our study. Although the mechanism of RNASE2 in gastric cancer has not been fully elucidated, prognostic analysis based on bioinformatics has provided important clues. A number of independent studies have consistently confirmed the potential value of RNASE2 as a prognostic biomarker for gastric cancer through large sample cohort analysis and multi-omics data mining [67–69]. It is worth noting that the latest research further reveals the gender-specific characteristics of the prognostic value of RNASE2. The study found that RNASE2 has a more prominent prognostic predictive ability in male patients, indicating that it may have a gender-dependent mechanism of action [66]. Therefore, RNASE2 has important clinical application potential in the prognosis evaluation of gastric cancer, especially in male patients, which may play a more critical role, and it is worth further studying its specific molecular mechanism.

In order to better understand how the tumor immune microenvironment differs between risk groups and identify potential mechanisms, we analyzed immune cell infiltration profiles in detail. Significant variations in the distribution of immune cell types were observed when comparing the high-risk and low-risk cohorts, and activated CD4 + T cell counts were considerably lower in the high-risk patient group, suggesting impaired adaptive immune responses in this subgroup. The DC can facilitate the activation of CD4 + T cells via signal transduction pathways, thereby enabling optimal initiation of the CD8 + T cell-mediated anti-tumor immune response [70,71], while the CD4 T cell exhaustion is associated with PD-1 expression and tumor immune escape [72,73]. Several recent studies have shown that through promoting lipid

peroxidation and ferroptosis, CD36 may promote CD8 T cell dysfunction and impair its anti-tumor ability [74,75]. On the other hand, mitochondrial fitness in Treg cells is regulated by CD36, which may contribute to their functional adaptation within a lactic acid-enriched tumor microenvironment, boosting their survival and immunosuppressive capabilities [76].

Through preliminary analysis, we have determined the prognostic significance of the risk model and its association with the immune status within the tumor microenvironment. In order to advance understanding the molecular biological mechanisms of different risk groups, we performed GSEA analysis and found that arrhythmogenic right ventricular cardiomyopathy (hsa05412) was significantly enriched in the high-risk group, and its molecular mechanism may also play a role in the initiation and progression of GC. Among them, Wnt/β-catenin signaling pathway contributes to key processes in the GC progression [77,78], and desmosome proteins, such as Plakophilin-2 (PKP2), can promote the activation of AKT/ mammalian target of rapamycin signaling pathway, thereby promoting the malignant progression of gastric cancer [79]. In addition, platelet activation (hsa04611) was also found to be significantly enriched in the high-risk group. Aggregated platelets can promote tumor growth by releasing pro-angiogenic mediators in the microvascular system [80]. Otherwise, platelets can also enhance endothelial retraction mediated by tumor cells and support their adhesion and invasion into the extracellular matrix, contributing to increased tumor cell proliferation and metastasis [81].

In order to explore personalized drug treatment strategies based on risk models, we conducted an in-depth analysis of drug sensitivity differences in different risk groups. The results showed that there were notable differences in the sensitivity of multiple antitumor drugs between HRG and LRG. The IC50 of BIBW2992 in HRG was higher than in LRG, indicating that high-risk patients showed stronger resistance to afatinib (BIBW2992). As a highly selective and irreversible erbB family inhibitor of tyrosine kinase receptors EGFR, HER2 and HER4, afatinib has shown a good therapeutic effect in HER2-positive GC [82]. Therefore, HRG patients may be accompanied by more complex gene mutation spectrum and signal pathway reprogramming, resulting in compensatory resistance to EGFR/HER2 inhibitors, thereby weakening the therapeutic effect of afatinib.

In this study, the expression characteristics of five prognostic genes in gastric cancer microenvironment were explored by single cell RNA sequencing analysis. The results showed that INHBA, ASCL2 and CD36 showed significant expression differences in dendritic cells, suggesting that these genes may be involved in the progression of gastric cancer by regulating the function of dendritic cells. From the perspective of mechanism, as a member of the TGF-β superfamily, the up-regulation of INHBA expression may affect the function of dendritic cells by activating the SMAD signaling pathway, which is known to be involved in the regulation of dendritic cells and promote gastric cancer [83–85]. ASCL2 is a downstream transcription factor of Wnt signaling pathway [84]. Its abnormal expression may affect the balance of immune microenvironment by regulating the maturation and antigen presentation function of dendritic cells [86]. Considering that dendritic cells are the most effective antigen presenting cells in tumor microenvironment [87], this regulation may be of great significance in immune escape of gastric cancer. As a fatty acid transporter [88], the expression of CD36 may reflect the change of cell metabolic state. Fatty acid metabolism not only has an important impact on the differentiation and homeostasis of dendritic cells, but also is closely related to the progression of gastric cancer [89]. In general, the synergistic changes of these genes may collectively affect the functional characteristics of dendritic cells in the tumor microenvironment, thereby regulating the anti-tumor immune response. However, there is also a notable phenomenon that the absolute expression levels of core prognostic genes such as INHBA and ASCL2 in dendritic cells are not high, although they show differences between cancer and normal tissues. This finding suggests that the mechanisms by which these tolDC - related prognostic genes affect the progression of gastric cancer may be more complex, and these predictive conclusions need to be further verified. However, single – cell level analysis can still enable us to locate the expression patterns of these genes in specific cell types. This provides a new perspective for understanding the regulatory mechanisms of the gastric cancer immune microenvironment, and at the same time, it also provides important clues for subsequent studies on the specific functions of these genes in dendritic cells.

Based on the transcriptome data set of GC patients and normal controls in the TCGA database, this study used machine learning methods to systematically mine tolDCs-RGs and their therapeutic targets in gastric cancer. Through bioinformatics methods, the function, immune characteristics, and potential regulatory mechanisms of these targets are deeply studied, and possible treatment strategies are explored, which provides an important reference for the clinical management of GC and the discovery of novel therapeutic agents. However, this study also has some limitations. Firstly, the predicted regulatory relationships of transcription factors have not been experimentally validated, reflecting a lack of analytical depth. Additionally, the findings are highly dependent on the quality of public databases and the accuracy of the algorithms employed, which may lead to false-positive or false-negative results. In subsequent work, we plan to expand the sample size and incorporate more detailed clinical data, combined with multi-omics analyses. Experimental approaches such as chromatin immunoprecipitation, gene knockout, gene editing, and animal models will be applied to verify the predicted regulatory mechanisms and further explore the molecular pathways involved. Future research should prioritize experimental validation and in-depth interpretation of results to enhance the reliability and practical application value of the findings.

## 5. Conclusion

This study focused on the association between tolerogenic dendritic cells (tolDCs) and gastric cancer (GC). Through screening, five genes with prognostic value—CXCL1, INHBA, ASCL2, RNASE1, and GPX3—were identified, and a risk model was constructed. The research suggested complex correlations between these prognostic genes and immune cells: GPX3 showed a significant positive correlation with various immune cell populations, particularly regulatory T cells, while ASCL2 had weak associations with almost all immune cells. Drug sensitivity analysis indicated that the high-risk group (HRG) had higher IC50 values for compounds such as BIBW2992, whereas the low-risk group (LRG) exhibited significantly higher IC50 values for drugs like GSK269962A. In summary, this study identified five tolDC-related prognostic genes for gastric cancer and established a predictive model. It not only provides a theoretical basis for understanding the correlation between tolDCs and gastric cancer but also lays the groundwork for formulating clinical treatment strategies and exploring new therapeutic targets for gastric cancer, thus holding important clinical reference value.

## Supporting information

**S1 Fig. Identification of differentially expressed genes related to tolDCs.** (**A-B**) DEGs distribution, volcano maps, and expression heat maps of tolDCs-DEGs2, and tolDCs-DEGs3. (**C**) Differentially expressed gene distribution volcano maps of tolDCs-DEGs4.
(TIF)

**S2 Fig. Single-cell analysis quality control and dimensionality reduction clustering.** (**A-B**) Single-cell filtering results showing gene counts (left), total mRNA molecules (middle), and mitochondrial gene percentage (right) (**A**) before and (**B**) after quality control. (**C**) Highly variable gene selection plot. The x-axis represents the average expression level of each gene in the sample, and the y-axis represents the degree of variation. Red dots indicate the 2,000 highly variable genes selected. (**D**) Two-dimensional PCA cell distribution plot. Different colors represent different samples, and each dot represents a single cell. (**E**) Principal component analysis results, highlighting the significant principal components. The significance of each principal component was assessed by comparing the p-value distribution of all genes on each principal component against a uniform distribution. (**F**) Scree plot of principal component analysis. The x-axis indicates the principal component dimension number, and the y-axis indicates the standard deviation. (**G**) UMAP cell distribution. The plot shows the cell distribution after UMAP dimensionality reduction, where the x-axis and y-axis represent the two main UMAP dimensions (UMAP1 and UMAP2), colored by cell clusters. (**H**) Dot plot of marker gene expression for each cell cluster before and after annotation. The x-axis shows different cell clusters, and the y-axis shows marker genes and their

corresponding cell types. (**I**) UMAP plot of cells after annotation. (**J**) Stacked bar chart showing different cell types across groups.
(TIF)

**S1 Table. 1003 tol DC-RGs.**
(XLSX)

**S2 Table. The GO analysis of candidate genes yielded a total of 510 BPs, 20 CCs, and 67 MFs (P < 0.05).**
(XLSX)

**S3 Table. 18 KEGG pathways with significant enrichment of candidate genes.**
(XLSX)

**S4 Table. All KEGG pathways enriched with prognostic genes.**
(XLSX)

## Acknowledgments

We would like to express our sincere gratitude to all individuals and organizations who supported and assisted us throughout this research. We extend our thanks to everyone who has supported and assisted us along the way. Without your support, this research would not have been possible.

## Author contributions

**Conceptualization:** Ying Liu, Shenyu Luo.

**Data curation:** Shenyu Luo.

**Formal analysis:** Guozhi Yang.

**Investigation:** Ying Liu, Guozhi Yang, Yong Zhang, Zhengrong Wen.

**Methodology:** Ying Liu, Guozhi Yang, Yuhua Yuan, Yong Zhang.

**Supervision:** Yihua Kang.

**Validation:** Yong Zhang, Yihua Kang, Wei Liu.

**Visualization:** Yihua Kang, Zhengrong Wen, Wei Liu.

**Writing – original draft:** Shenyu Luo.

**Writing – review & editing:** Ying Liu, Shenyu Luo, Yuhua Yuan.

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
