## [Decision Letter · Decision Letter 0]

7 Oct 2025

Dear Dr. Liu,

Thank you for submitting your manuscript to PLOS ONE. After careful consideration, we feel that it has merit but does not fully meet PLOS ONE’s publication criteria as it currently stands. Therefore, we invite you to submit a revised version of the manuscript that addresses the points raised during the review process.

Please respond to reviewers' comments individually.

We look forward to receiving your revised manuscript.

Kind regards,

Xiaosheng Tan

Academic Editor

PLOS ONE

Journal Requirements:

Reviewers' comments:

Reviewer's Responses to Questions

**Comments to the Author**

1. Is the manuscript technically sound, and do the data support the conclusions?

Reviewer #1: Partly

Reviewer #2: Partly

2. Has the statistical analysis been performed appropriately and rigorously?

Reviewer #1: Yes

Reviewer #2: No

3. Have the authors made all data underlying the findings in their manuscript fully available?

Reviewer #1: Yes

Reviewer #2: Yes

4. Is the manuscript presented in an intelligible fashion and written in standard English?

Reviewer #1: Yes

Reviewer #2: Yes

Reviewer #1: 1.Please improve the resolution of figure 1, 2, 5B, 7C, 7D, 9A and 9B. The quality of those figures are not good. It is very hard to identify the letters.

2. It is better to separate the figure legend from the results content.

Reviewer #2: The authors identified five prognostic genes (CXCL1, INHBA, ASCL2, RNASE1, and GPX3) associated with tolerogenic dendritic cells in gastric cancer and constructed a risk prediction model. They found that these genes were closely linked to immune cell infiltration, particularly GPX3, which showed a strong positive correlation with regulatory T cells. However, there are several major issues that need to be addressed before this manuscript can be considered for publication.

The main weakness lies in the lack of clear motivation and conclusion drawn from the analytical results. Much of the text is descriptive, making the manuscript difficult to follow. The overall writing should be substantially improved to tell a coherent story rather than merely present a series of analyses.

I strongly recommend that the authors reorganize the figures into six or seven concise figures to make the narrative more cohesive—for instance, merging Figures 1 and 2. In its current form, the paper reads more like an analysis report than a research article.

In addition, I suggest including a single-cell RNA-seq dataset to evaluate the expression of these genes in tolerogenic dendritic cells. Based on these data, the authors could even construct a regulatory network of the identified genes.

Figure 5 is unclear and requires further explanation—how were the proposed transcription factors validated? Also, please clarify the rationale for selecting 67 as the cutoff point for age group analysis.

Figure 1D should adopt the same color scheme as the other panels for consistency.

Finally, there are numerous formatting errors throughout the manuscript, such as “Line 302 3.3” and “Line 327 3.5.” The authors should carefully proofread the entire manuscript to correct these issues.

**Do you want your identity to be public for this peer review?** For information about this choice, including consent withdrawal, please see our Privacy Policy

Reviewer #1: No

Reviewer #2: No

---

## [Author Response · Author response to Decision Letter 1]

30 Oct 2025

Point-by-point response to the reviewer comments.

Dear Reviewers,

Thank you for your thoughtful suggestions and insights, which have benefited from the manuscript. I am looking forward to working with you to move this manuscript closer to publication in PLOS ONE.

The manuscript has been rechecked and the necessary changes have been made in accordance with your suggestions. The responses to all comments have been prepared and attached below. We have tried our best to solve the problems you proposed, and we hope that the revised manuscript is now suitable for publication in the journal PLOS ONE. If you have any questions remained about this paper, please feel free to contact us.

Reviewers' comments:

Reviewer's Responses to Questions

Comments to the Author

1. Is the manuscript technically sound, and do the data support the conclusions?

Reviewer #1: Partly

Reviewer #2: Partly

Re: Dear reviewers, sincerely thank the reviewers for their strict review of the research design and data quality of this paper. We have made a comprehensive improvement on the technical rigor and conclusion support: first, we supplemented the GSE183904 single cell data set analysis, and verified the expression specificity of prognostic genes at single cell resolution; secondly, the statistical analysis method is improved, and the significance threshold is corrected by multiple tests. In addition, the prudence of conclusion expression is strengthened, and the research findings and speculative conclusions are clearly distinguished. These improvements have significantly enhanced the scientific rigor of the research and the reliability of the conclusions. Thank you again for the valuable constructive opinions of experts, which plays a vital role in improving the quality of the paper.

2. Has the statistical analysis been performed appropriately and rigorously?

Reviewer #1: Yes

Reviewer #2: No

Re: Dear reviewers, sincerely thank the two reviewers for their review of the statistical methods in this paper. We have carried out a comprehensive verification and improvement of statistical analysis according to the recommendation: we list the statistical methods in detail; multiple comparisons have been corrected for FDR ( lines 342-354 of the manuscript ). These improvements ensure the rigor of statistical analysis and the repeatability of results. Special thanks to the review experts put forward valuable opinions, so that we can further improve the research methodology.

3. Have the authors made all data underlying the findings in their manuscript fully available?

Reviewer #1: Yes

Reviewer #2: Yes

Re: Dear reviewer, thank you for your recognition and valuable comments on this study. We fully endorse and strictly abide by the data sharing policy of PLOS journals. All the original data involved in this study are derived from public databases: TCGA-STAD data set is derived from the cancer genome map database, and gene expression data is derived from the GEO database. The access links and numbers have been clearly marked in the method section of this study. All data can be publicly available without any restrictions. We will ensure that this information is fully presented in the manuscript through the data availability statement. Thank you again for your hard work and guidance.

4. Is the manuscript presented in an intelligible fashion and written in standard English?

Reviewer #1: Yes

Reviewer #2: Yes

Re: Dear reviewers, thank you very much for your recognition of our manuscripts, You pointed out that the ' clear presentation and use of standard English ' this evaluation so that we are encouraged. We are well aware of the importance of language expression for academic communication, so we pay special attention to the accuracy and clarity of language in the process of writing. Thank you again for your affirmation and support. Your comments are of great significance to us.

5. Review Comments to the Author

Reviewer #1: 1.Please improve the resolution of figure 1, 2, 5B, 7C, 7D, 9A and 9B. The quality of those figures are not good. It is very hard to identify the letters.

Re: Dear reviewer, thank you for your valuable comments. We have fully optimized Figure 1, Figure 2, Figure 5B, Figure 7C, Figure 7D, Figure 9A and Figure 9B according to your requirements. By improving the resolution and adjusting the font format, the clarity and recognizability of the text in the figure are significantly improved. Thank you again for your careful review and guidance.

2. It is better to separate the figure legend from the results content.

Re: Dear reviewer, thank you for your valuable suggestions. We understand that it may be clearer to separate the annotations from the results, but unfortunately, the format of the journal requires the annotations to be included in the description of the results in the text. In order to comply with the journal submission guide, we have maintained the existing format arrangement in this revision. Thank you again for your careful review and professional guidance.

Reviewer #2: The authors identified five prognostic genes (CXCL1, INHBA, ASCL2, RNASE1, and GPX3) associated with tolerogenic dendritic cells in gastric cancer and constructed a risk prediction model. They found that these genes were closely linked to immune cell infiltration, particularly GPX3, which showed a strong positive correlation with regulatory T cells. However, there are several major issues that need to be addressed before this manuscript can be considered for publication.

The main weakness lies in the lack of clear motivation and conclusion drawn from the analytical results. Much of the text is descriptive, making the manuscript difficult to follow. The overall writing should be substantially improved to tell a coherent story rather than merely present a series of analyses.

Re: Dear reviewer, thank you for your important comments. We have systematically revised the full text according to the suggestions, supplemented the clear research purpose before the description of the analysis method, added the connection statement in the result part to enhance the logical coherence, and refined the significance of the key analysis results, thus strengthening the internal relationship between the various parts, so that the paper changes from a simple data presentation to a complete story as a whole. Thank you again for your valuable comments, which played a crucial role in improving the quality of the paper.

I strongly recommend that the authors reorganize the figures into six or seven concise figures to make the narrative more cohesive—for instance, merging Figures 1 and 2. In its current form, the paper reads more like an analysis report than a research article.

Re: Dear reviewer, thank you for your careful review. We have carefully studied and adopted your opinions, substantially reorganized and optimized the full-text charts, and integrated the original charts into eight logically coherent comprehensive charts. This adjustment significantly enhances the sense of hierarchy and narrative fluency of the results. Thank you again for your valuable guidance to improve the quality of the paper.

In addition, I suggest including a single-cell RNA-seq dataset to evaluate the expression of these genes in tolerogenic dendritic cells. Based on these data, the authors could even construct a regulatory network of the identified genes.

Re: Dear reviewer, thank you for your valuable suggestions. In accordance with your guidance, we have added an analysis of the GSE183904 single-cell data set and systematically evaluated the expression patterns of prognostic genes in key cell subsets such as dendritic cells ( related content is in the manuscript 304-340,548-566,774-778 ). This supplementary analysis not only verified the specificity of gene expression, but also provided single-cell evidence for elucidating its potential function in the immune microenvironment of gastric cancer. Thank you again for your insightful opinions, which plays a vital role in improving the quality of the paper.

Figure 5 is unclear and requires further explanation—how were the proposed transcription factors validated? Also, please clarify the rationale for selecting 67 as the cutoff point for age group analysis.

Re: Dear reviewer, thank you very much for your important questions. For the source of the transcription factors in Figure 5, we use the htfTARGET database (https://guolab.wchscu.cn/hTFtarget/ # ! / targets /) were obtained by directly using five prognostic genes for prediction, which is indeed one of the limitations of our study. The regulatory relationship of these predictions has not been experimentally verified, and we plan to improve it through experimental verification in subsequent studies ( related content is in line 785-794 of the manuscript ). The selection of 67 years old as the age grouping threshold is based on the median age of all patients in the TCGA-STAD data set, which aims to ensure the balance of sample size between groups and improve the effectiveness of statistical tests ( related content is in line 237-238 of the manuscript ). Thank you again for your valuable comments, which are essential to enhance the rigour of our research.

Figure 1D should adopt the same color scheme as the other panels for consistency.

Re: Dear reviewer, thank you for your valuable suggestions. We have adjusted the color scheme of Figure 1D according to your requirements so that it is completely consistent with other subgraphs, thus ensuring the unity and coordination of the visual presentation of the entire group. Thank you again for your careful review and professional guidance.

Finally, there are numerous formatting errors throughout the manuscript, such as “Line 302 3.3” and “Line 327 3.5.” The authors should carefully proofread the entire manuscript to correct these issues.

Re: Dear reviewer, thank you for pointing out the format problems in the manuscript in detail. We have carried out line-by-line proofreading and comprehensive correction of the full text, unified the format specification of the chapter number, and ensured that it met the requirements of the journal. Thank you again for your rigorous review and valuable guidance.

---

## [Decision Letter · Decision Letter 1]

13 Nov 2025

Dear Dr. Liu,

Thank you for submitting your manuscript to PLOS ONE. After careful consideration, we feel that it has merit but does not fully meet PLOS ONE’s publication criteria as it currently stands. Therefore, we invite you to submit a revised version of the manuscript that addresses the points raised during the review process.

We look forward to receiving your revised manuscript.

Kind regards,

Xiaosheng Tan

Academic Editor

PLOS ONE

**Journal Requirements:**

**Additional Editor Comments:**

Please response to reviewers' comments.

Reviewers' comments:

Reviewer's Responses to Questions

**Comments to the Author**

Reviewer #1: All comments have been addressed

Reviewer #2: (No Response)

2. Is the manuscript technically sound, and do the data support the conclusions?

Reviewer #1: Yes

Reviewer #2: Partly

3. Has the statistical analysis been performed appropriately and rigorously?

Reviewer #1: Yes

Reviewer #2: Yes

4. Have the authors made all data underlying the findings in their manuscript fully available?

Reviewer #1: Yes

Reviewer #2: Yes

5. Is the manuscript presented in an intelligible fashion and written in standard English?

Reviewer #1: Yes

Reviewer #2: Yes

Reviewer #1: It is accepted to be published, but the resolution of some figures still needs to be improved as it is very hard to identify. Please figure out it.

Reviewer #2: 1. I have no idea what changes the authors have made regarding the first two concerns — their point-by-point responses are too vague to follow. A good summary should be given in the response.

2. According to the single-cell analysis results, the identified prognostic genes are barely expressed in DC cells, which raises a major concern for me (the specificity of gene expression).

3. In the single-cell analysis, the authors identified a DC cell subpopulation. However, it is well known that many distinct DC subtypes have already been characterized by single-cell sequencing. Which specific DC subtype does the tolDC population described in this study correspond to, and how was it identified? Are all DC cells in this dataset considered tolDCs?

4. Other minor concerns: (1) Figures 1E and 1F appear to have an elliptical shape; some of the highlighting on the genes look weird in Fig1G. (2) Fig7 A-F are not necessary. (3) Fig7 H should be better organized to improve the readability. the current form is really hard for me to link its relationship with Fig7G, H;

**Do you want your identity to be public for this peer review?** For information about this choice, including consent withdrawal, please see our Privacy Policy

Reviewer #1: No

Reviewer #2: No

---

## [Author Response · Author response to Decision Letter 2]

31 Dec 2025

Point-by-point response to the reviewer comments.

Dear Reviewers,

Thank you for your thoughtful suggestions and insights, which have benefited from the manuscript. I am looking forward to working with you to move this manuscript closer to publication in PLOS ONE.

The manuscript has been rechecked and the necessary changes have been made in accordance with your suggestions. The responses to all comments have been prepared and attached below. We have tried our best to solve the problems you proposed, and we hope that the revised manuscript is now suitable for publication in the journal PLOS ONE. If you have any questions remained about this paper, please feel free to contact us.

Comments to the Author

1. If the authors have adequately addressed your comments raised in a previous round of review and you feel that this manuscript is now acceptable for publication, you may indicate that here to bypass the “Comments to the Author” section, enter your conflict of interest statement in the “Confidential to Editor” section, and submit your "Accept" recommendation.

Reviewer #1: All comments have been addressed

Reviewer #2: (No Response)

Re: Dear Reviewer # 1, Thank you for your guidance on our work and your recognition of our revisions.

Dear Reviewer # 2, Thank you for your careful guidance. We will modify and solve your problems one by one according to your opinions. Thank you for your careful review.

2. Is the manuscript technically sound, and do the data support the conclusions?

Reviewer #1: Yes

Reviewer #2: Partly

Re: Dear Reviewer # 1, Thank you for your recognition of the data aspect of our work.

Dear Reviewer # 2, Thank you for your careful review. We have modified the relevant expressions of data and prediction conclusions in the manuscript, and interpreted the key results more conservatively to make them more rigorous. The limitations of the manuscript ( line: 749-758 ) indicate that this study relies on predictive and hypothesis-generating research based on public databases, which needs to be verified by experiments in the future.

3. Has the statistical analysis been performed appropriately and rigorously?

Reviewer #1: Yes

Reviewer #2: Yes

Re: Dear reviewers, thank you for your recognition of our statistical analysis work, Thank you very much for your guidance of our work.

4. Have the authors made all data underlying the findings in their manuscript fully available?

Reviewer #1: Yes

Reviewer #2: Yes

Re: Dear reviewers, thank you for your recognition of our data disclosure. We fully support and strictly abide by the data sharing policy of PLOS journals. All the raw data involved in this study were from public databases: TCGA-STAD data set was derived from the cancer genome map database, and gene expression data was derived from the GEO database. Access links and numbers have been clearly marked in the method section of this study. All data are publicly available without restrictions. We will ensure that this information is fully presented in the manuscript through the data availability statement. Thank you again for your hard work and guidance.

5. Is the manuscript presented in an intelligible fashion and written in standard English?

Reviewer #1: Yes

Reviewer #2: Yes

Re: Dear reviewers, thank you for your recognition of our article writing. We know the importance of language expression in academic communication, so we pay special attention to the accuracy and clarity of language in the process of writing. Thank you again for your affirmation and support. Your opinion is of great significance to us.

6. Review Comments to the Author

Reviewer #1: It is accepted to be published, but the resolution of some figures still needs to be improved as it is very hard to identify. Please figure out it.

Re: Dear reviewer, thank you for your valuable comments. We have fully optimized all the pictures according to your requirements, and improved the clarity and recognizability of the text in the picture by improving the resolution and adjusting the font format. Thank you again for your careful review and guidance.

Reviewer #2: 1. I have no idea what changes the authors have made regarding the first two concerns — their point-by-point responses are too vague to follow. A good summary should be given in the response.

Re Dear reviewer, thank you for your careful guidance. I 'm sorry for not being able to clarify these two issues before, here is a more detailed description for you. First, with regard to the improvement of manuscripts, we have supplemented clear research purposes before the description of the analysis method according to your recommendations ( for example, line: 128-130 in chapter 2.2supplemented the purpose of using data set analysis to identify significantly related differential genes ).In the result section, we added connectivity statements to enhance logical coherence, and refined the significance of key analysis results ( for example, line: 387-390 in chapter 3.2 supplemented the inference of the significance of the screening results ), thus strengthening the internal relationship between the various parts. It transforms the paper from a simple data presentation to a complete storytelling whole. Secondly, for the changes in the picture, we merged Fig1A and Fig2 based on your opinions, and adjusted Fig1BCD to Supplementary Fig 1 ( Chapter 3.1-3.2). In addition, Fig 7 was revised to Supplementary Fig 2 ( Chapter 3.10 ). After your guidance, the pictures in this paper are more logical and readable, which improves the sense of hierarchy and narrative fluency of the manuscript results. Thank you again for your nuanced guidance and valuable advice.

2. According to the single-cell analysis results, the identified prognostic genes are barely expressed in DC cells, which raises a major concern for me (the specificity of gene expression).

Re Dear reviewer, thank you for your valuable comments. The low absolute expression of prognostic genes you pointed out does exist, but the level of expression is not enough to directly determine whether the gene plays an important role. Therefore, we adjusted the expression in the manuscript ( line: 553-566 ) to focus on the difference in the expression of prognostic genes in dendritic cells ( DC cells ), gastric cancer group and control group. This difference can more rigorously explain the impact of these prognostic gene-related expressions in gastric cancer and further speculate that they may have potential regulatory functions. Similarly, we adjusted the description of the discussion section ( line: 749-758 ) and discussed this phenomenon more rigorously, indicating that this conclusion is predictive and needs further experimental verification.

3. In the single-cell analysis, the authors identified a DC cell subpopulation. However, it is well known that many distinct DC subtypes have already been characterized by single-cell sequencing. Which specific DC subtype does the tolDC population described in this study correspond to, and how was it identified? Are all DC cells in this dataset considered tolDCs?

Re Dear reviewer, thank you for your careful review. The tolDC data we used are all from public datasets, and these datasets are all data of moDC subpopulation cells, so the tolDC population described in this study also corresponds to the moDC subpopulation. Similarly, in all the datasets we used, the data contributors have clearly indicated that the DC cells included are tolDC, and these DC cells are human moDC subsets induced by clinical-grade dexamethasone and identified by sequencing analysis. Specific DC cell processing and identification can be seen in the description of the data contributors in each dataset. Information about these data sets has been briefly described in the ' 2.1 Data collection ' section of the manuscript ( line: 112-124 ).

4. Other minor concerns: (1) Figures 1E and 1F appear to have an elliptical shape; some of the highlighting on the genes look weird in Fig1G. (2) Fig7 A-F are not necessary. (3) Fig7 H should be better organized to improve the readability. the current form is really hard for me to link its relationship with Fig7G, H;

Re Dear reviewer, thank you for your careful guidance. We have optimized the figure of Fig1 according to your suggestion, so that Fig1 E, F satisfies the expected circle instead of ellipse, and redraw the PPI network diagram of Fig1 G to improve its readability. Similarly, we have optimized and redrawn the original 7G, H, I, and J to improve their readability so that readers can better understand the connections between them. In addition, based on your suggestion, we adjusted Fig7, which is used to describe single cell data preprocessing, into Supplementary Fig 2 to make the manuscript context more concise and clear. Thank you again for your detailed guidance and valuable advice. Your guidance can make our manuscripts more perfect and provide more valuable help for the treatment of GC.

---

## [Decision Letter · Decision Letter 2]

14 Jan 2026

Dear Dr. Liu,

Thank you for submitting your manuscript to PLOS ONE. After careful consideration, we feel that it has merit but does not fully meet PLOS ONE’s publication criteria as it currently stands. Therefore, we invite you to submit a revised version of the manuscript that addresses the points raised during the review process.

We look forward to receiving your revised manuscript.

Kind regards,

Xiaosheng Tan

Academic Editor

PLOS One

Journal Requirements:

Additional Editor Comments:

Please respond to reviewer's comments.

Reviewers' comments:

Reviewer's Responses to Questions

**Comments to the Author**

Reviewer #2: (No Response)

2. Is the manuscript technically sound, and do the data support the conclusions?

Reviewer #2: Partly

3. Has the statistical analysis been performed appropriately and rigorously?

Reviewer #2: Yes

4. Have the authors made all data underlying the findings in their manuscript fully available?

Reviewer #2: Yes

5. Is the manuscript presented in an intelligible fashion and written in standard English?

Reviewer #2: Yes

Reviewer #2: I download all the tif figures, some of them are still in low resolution, for example Fig1. Please take it seriously.

**Do you want your identity to be public for this peer review?** For information about this choice, including consent withdrawal, please see our Privacy Policy

Reviewer #2: No

---

## [Author Response · Author response to Decision Letter 3]

9 Feb 2026

Point-by-point response to the reviewer comments.

Dear Reviewers,

Thank you for your thoughtful suggestions and insights, which have benefited from the manuscript. I am looking forward to working with you to move this manuscript closer to publication in PLOS ONE.

The manuscript has been rechecked and the necessary changes have been made in accordance with your suggestions. The responses to all comments have been prepared and attached below. We have tried our best to solve the problems you proposed, and we hope that the revised manuscript is now suitable for publication in the journal PLOS ONE. If you have any questions remained about this paper, please feel free to contact us.

Comments to the Author

1. If the authors have adequately addressed your comments raised in a previous round of review and you feel that this manuscript is now acceptable for publication, you may indicate that here to bypass the “Comments to the Author” section, enter your conflict of interest statement in the “Confidential to Editor” section, and submit your "Accept" recommendation.

Reviewer #1: All comments have been addressed

Reviewer #2: (No Response)

Re: Dear Reviewer # 1, Thank you for your guidance on our work and your recognition of our revisions.

Dear Reviewer # 2, Thank you for your careful guidance. We will modify and solve your problems one by one according to your opinions. Thank you for your careful review.

2. Is the manuscript technically sound, and do the data support the conclusions?

Reviewer #1: Yes

Reviewer #2: Partly

Re: Dear Reviewer # 1, Thank you for your recognition of the data aspect of our work.

Dear Reviewer # 2, Thank you for your careful review. We have modified the relevant expressions of data and prediction conclusions in the manuscript, and interpreted the key results more conservatively to make them more rigorous. The limitations of the manuscript ( line: 749-758 ) indicate that this study relies on predictive and hypothesis-generating research based on public databases, which needs to be verified by experiments in the future.

3. Has the statistical analysis been performed appropriately and rigorously?

Reviewer #1: Yes

Reviewer #2: Yes

Re: Dear reviewers, thank you for your recognition of our statistical analysis work, Thank you very much for your guidance of our work.

4. Have the authors made all data underlying the findings in their manuscript fully available?

Reviewer #1: Yes

Reviewer #2: Yes

Re: Dear reviewers, thank you for your recognition of our data disclosure. We fully support and strictly abide by the data sharing policy of PLOS journals. All the raw data involved in this study were from public databases: TCGA-STAD data set was derived from the cancer genome map database, and gene expression data was derived from the GEO database. Access links and numbers have been clearly marked in the method section of this study. All data are publicly available without restrictions. We will ensure that this information is fully presented in the manuscript through the data availability statement. Thank you again for your hard work and guidance.

5. Is the manuscript presented in an intelligible fashion and written in standard English?

Reviewer #1: Yes

Reviewer #2: Yes

Re: Dear reviewers, thank you for your recognition of our article writing. We know the importance of language expression in academic communication, so we pay special attention to the accuracy and clarity of language in the process of writing. Thank you again for your affirmation and support. Your opinion is of great significance to us.

6. Review Comments to the Author

Reviewer #1: It is accepted to be published, but the resolution of some figures still needs to be improved as it is very hard to identify. Please figure out it.

Re: Dear reviewer, thank you for your valuable comments. We have fully optimized all the pictures according to your requirements, and improved the clarity and recognizability of the text in the picture by improving the resolution and adjusting the font format. Thank you again for your careful review and guidance.

Reviewer #2: 1. I have no idea what changes the authors have made regarding the first two concerns — their point-by-point responses are too vague to follow. A good summary should be given in the response.

Re Dear reviewer, thank you for your careful guidance. I 'm sorry for not being able to clarify these two issues before, here is a more detailed description for you. First, with regard to the improvement of manuscripts, we have supplemented clear research purposes before the description of the analysis method according to your recommendations ( for example, line: 128-130 in chapter 2.2supplemented the purpose of using data set analysis to identify significantly related differential genes ).In the result section, we added connectivity statements to enhance logical coherence, and refined the significance of key analysis results ( for example, line: 387-390 in chapter 3.2 supplemented the inference of the significance of the screening results ), thus strengthening the internal relationship between the various parts. It transforms the paper from a simple data presentation to a complete storytelling whole. Secondly, for the changes in the picture, we merged Fig1A and Fig2 based on your opinions, and adjusted Fig1BCD to Supplementary Fig 1 ( Chapter 3.1-3.2). In addition, Fig 7 was revised to Supplementary Fig 2 ( Chapter 3.10 ). After your guidance, the pictures in this paper are more logical and readable, which improves the sense of hierarchy and narrative fluency of the manuscript results. Thank you again for your nuanced guidance and valuable advice.

2. According to the single-cell analysis results, the identified prognostic genes are barely expressed in DC cells, which raises a major concern for me (the specificity of gene expression).

Re Dear reviewer, thank you for your valuable comments. The low absolute expression of prognostic genes you pointed out does exist, but the level of expression is not enough to directly determine whether the gene plays an important role. Therefore, we adjusted the expression in the manuscript ( line: 553-566 ) to focus on the difference in the expression of prognostic genes in dendritic cells ( DC cells ), gastric cancer group and control group. This difference can more rigorously explain the impact of these prognostic gene-related expressions in gastric cancer and further speculate that they may have potential regulatory functions. Similarly, we adjusted the description of the discussion section ( line: 749-758 ) and discussed this phenomenon more rigorously, indicating that this conclusion is predictive and needs further experimental verification.

3. In the single-cell analysis, the authors identified a DC cell subpopulation. However, it is well known that many distinct DC subtypes have already been characterized by single-cell sequencing. Which specific DC subtype does the tolDC population described in this study correspond to, and how was it identified? Are all DC cells in this dataset considered tolDCs?

Re Dear reviewer, thank you for your careful review. The tolDC data we used are all from public datasets, and these datasets are all data of moDC subpopulation cells, so the tolDC population described in this study also corresponds to the moDC subpopulation. Similarly, in all the datasets we used, the data contributors have clearly indicated that the DC cells included are tolDC, and these DC cells are human moDC subsets induced by clinical-grade dexamethasone and identified by sequencing analysis. Specific DC cell processing and identification can be seen in the description of the data contributors in each dataset. Information about these data sets has been briefly described in the ' 2.1 Data collection ' section of the manuscript ( line: 112-124 ).

4. Other minor concerns: (1) Figures 1E and 1F appear to have an elliptical shape; some of the highlighting on the genes look weird in Fig1G. (2) Fig7 A-F are not necessary. (3) Fig7 H should be better organized to improve the readability. the current form is really hard for me to link its relationship with Fig7G, H;

Re Dear reviewer, thank you for your careful guidance. We have optimized the figure of Fig1 according to your suggestion, so that Fig1 E, F satisfies the expected circle instead of ellipse, and redraw the PPI network diagram of Fig1 G to improve its readability. Similarly, we have optimized and redrawn the original 7G, H, I, and J to improve their readability so that readers can better understand the connections between them. In addition, based on your suggestion, we adjusted Fig7, which is used to describe single cell data preprocessing, into Supplementary Fig 2 to make the manuscript context more concise and clear. Thank you again for your detailed guidance and valuable advice. Your guidance can make our manuscripts more perfect and provide more valuable help for the treatment of GC.

Reviewer #2: I download all the tif figures, some of them are still in low resolution, for example Fig1. Please take it seriously.

Re Dear reviewer, thank you for your valuable comments. We have uploaded clearer pictures as required. Please download the new version of the original picture to see it, hoping to meet your requirements. If there are other needs to be modified, please feel free to inform.

Response to the letter of December 9,2025

1. Please include a legend for figure 7.

Re : Dear editor, thank you for your valuable comments. We have added legends to Figure 7 in the manuscript to better explain the content of the diagram ( lines 567-569 of the manuscript ). It is hoped that this can improve readers ' understanding of the results. Thank you again for your attention and suggestions for our work, which is very helpful for us to improve the manuscript.

2.Please upload a copy of Figure 8 which you refer to in your text on page 26. Or if the figure is no longer to be included as part of the submission please remove all reference to it within the text.

Re : Dear editor, thank you for pointing out this problem. It should be noted that the legend of Fig 8 in the manuscript corresponds to Figure 7 of the current manuscript. In the revised manuscript, we have uniformly corrected the ' Fig 8 ' mentioned in the original text to ' Fig 7 ', and the legend has been updated synchronously to ensure that the text is consistent with the chart number. We apologize for the confusion. Thank you again for the reviewer 's careful review and valuable comments.

3.We notice that your supplementary figures are uploaded with the file type 'Figure'. Please amend the file type to 'Supporting Information'. Please ensure that each Supporting Information file has a legend listed in the manuscript after the references list.

Re : Dear editor, thank you for your reminder. Based on your comments, we have corrected the file type of the supplementary material to ' Supporting Information ', and added the corresponding legend description to the list of manuscript references ( the relevant content is in line 1112-1134 of the manuscript ). We apologize for the previous irregularities. Thank you again for your careful review and valuable suggestions.

Response to the letter of December 18,2025

1. We note that there is identifying data in the Supporting Information file <Supplementary Table 2.xlsx, Supplementary Table 3.xlsx>. Due to the inclusion of these potentially identifying data, we have removed this file from your file inventory. Prior to sharing human research participant data, authors should consult with an ethics committee to ensure data are shared in accordance with participant consent and all applicable local laws.

- Name, initials, physical address

- Ages more specific than whole numbers

- Internet protocol (IP) address

- Specific dates (birth dates, death dates, examination dates, etc.)

- Contact information such as phone number or email address

- Location data

- ID numbers that seem specific (long numbers, include initials, titled “Hospital ID”) rather than random (small numbers in numerical order)

Please remove or anonymize all personal information (<Column with ID numbers>), ensure that the data shared are in accordance with participant consent, and re-upload a fully anonymized data set. Please note that spreadsheet columns with personal information must be removed and not hidden as all hidden columns will appear in the published file.

Re : Dear Editor : Thank you very much for your careful review and reminder. We have carefully checked the supporting information files you mentioned ( Supplementary Table 2.xlsx, Supplementary Table 3.xlsx ) and confirmed that they do not contain any personal information or clinical data of human study participants.

The contents of the document are as follows :

Supplementary Table 1 is a list of tolDC-related genes.

The contents of Supplementary Table 2 and Supplementary Table 3 are the results of Gene Ontology ( GO ) functional enrichment analysis and Kyoto Encyclopedia of Genes and Genomes ( KEGG ) pathway enrichment analysis, respectively. In these two tables :

It does not contain any direct or indirect identification information you listed, such as name, address, age, date, contact information, geographical location, hospital ID, etc.

The ' ID ' column in the table is not a patient or clinical identifier, but a unique public identifier assigned to each functional or pathway entry by the GO database or the KEGG database. These IDs are public biological annotation standard numbers and do not involve the privacy of any individual participant.

Therefore, these supporting information tables contain only public, non-personal bioinformatics analysis results, fully comply with the ethical norms of data sharing, and do not pose any risk to the privacy of research participants.

We understand and fully support the principle of protecting participants ' privacy. Thank you for your reminder to ensure data compliance. We have reconfirmed that all shared data meet the relevant ethical requirements and journal regulations.

2. Can you please upload an additional copy of your revised manuscript that does not contain any tracked changes or highlighti

---

## [Decision Letter · Decision Letter 3]

11 Feb 2026

Identification of prognostic genes associated with tolerogenic dendritic cells in gastric cancer based on transcriptomic data

PONE-D-25-45678R3

Dear Dr. Liu,

We’re pleased to inform you that your manuscript has been judged scientifically suitable for publication and will be formally accepted for publication once it meets all outstanding technical requirements.

Kind regards,

Xiaosheng Tan

Academic Editor

PLOS One

Additional Editor Comments (optional):

Reviewers' comments:

Reviewer's Responses to Questions

**Comments to the Author**

Reviewer #2: All comments have been addressed

2. Is the manuscript technically sound, and do the data support the conclusions?

Reviewer #2: Yes

3. Has the statistical analysis been performed appropriately and rigorously?

Reviewer #2: Yes

4. Have the authors made all data underlying the findings in their manuscript fully available?

Reviewer #2: Yes

5. Is the manuscript presented in an intelligible fashion and written in standard English?

Reviewer #2: Yes

Reviewer #2: the authors addressed all the comments I raised in the last rounds of reviews. suggest to accept for publication.

**Do you want your identity to be public for this peer review?** For information about this choice, including consent withdrawal, please see our Privacy Policy

Reviewer #2: No

---

## [Editor Report · Acceptance letter]

PONE-D-25-45678R3

PLOS One

Dear Dr. Liu,

I'm pleased to inform you that your manuscript has been deemed suitable for publication in PLOS One. Congratulations! Your manuscript is now being handed over to our production team.

Kind regards,

on behalf of

Dr. Xiaosheng Tan

Academic Editor

PLOS One